# The EMT activator ZEB1 accelerates endosomal trafficking to establish a polarity axis in lung adenocarcinoma cells

Priyam Banerjee[1,8,9], Guan-Yu Xiao[1,9], Xiaochao Tan[1], Veronica J. Zheng[1], Lei Shi[1],
Maria Neus Bota Rabassedas [2], Hou-fu Guo[3], Xin Liu[1], Jiang Yu[1], Lixia Diao [4], Jing Wang [4],
William K. Russell [5], Jason Roszik [6], Chad J. Creighton [7] & Jonathan M. Kurie [1✉]

Epithelial-to-mesenchymal transition (EMT) is a transcriptionally governed process by which cancer cells establish a front-rear polarity axis that facilitates motility and invasion. Dynamic assembly of focal adhesions and other actin-based cytoskeletal structures on the leading edge of motile cells requires precise spatial and temporal control of protein trafficking. Yet, the way in which EMT-activating transcriptional programs interface with vesicular trafficking networks that effect cell polarity change remains unclear. Here, by utilizing multiple approaches to assess vesicular transport dynamics through endocytic recycling and retrograde trafficking pathways in lung adenocarcinoma cells at distinct positions on the EMT spectrum, we find that the EMT-activating transcription factor ZEB1 accelerates endocytosis and intracellular trafficking of plasma membrane-bound proteins. ZEB1 drives turnover of the MET receptor tyrosine kinase by hastening receptor endocytosis and transport to the lysosomal compartment for degradation. ZEB1 relieves a plus-end-directed microtubule-dependent kinesin motor protein (KIF13A) and a clathrin-associated adaptor protein complex subunit (AP1S2) from microRNA-dependent silencing, thereby accelerating cargo transport through the endocytic recycling and retrograde vesicular pathways, respectively. Depletion of KIF13A or AP1S2 mitigates ZEB1-dependent focal adhesion dynamics, front-rear axis polarization, and cancer cell motility. Thus, ZEB1-dependent transcriptional networks govern vesicular trafficking dynamics to effect cell polarity change.

[1] Department of Thoracic/Head and Neck Medical Oncology, The University of Texas MD Anderson Cancer Center, Houston, TX, USA. [2] Department of Translational Molecular Pathology, The University of Texas MD Anderson Cancer Center, Houston, TX, USA. [3] Department of Molecular and Cellular Biochemistry, University of Kentucky, Lexington, KY, USA. [4] Department of Bioinformatics and Computational Biology, The University of Texas MD Anderson Cancer Center, Houston, TX, USA. [5] Department of Biochemistry and Molecular Biology, University of Texas Medical Branch, Galveston, TX, USA. [6] Department of Melanoma Medical Oncology, The University of Texas MD Anderson Cancer Center, Houston, TX, USA. [7] Department of Medicine, Dan L. Duncan Cancer Center, Baylor College of Medicine, Houston, TX, USA. [8] Present address: Bio-Imaging Resource Center, The Rockefeller University, New York, NY, USA. [9] These authors contributed equally: Priyam Banerjee, Guan-Yu Xiao. ✉email: jkurie@mdanderson.org

Metastatic disease is a poor prognostic feature and the primary cause of death in patients with epithelial cancers[1]. Cancer cells detach from the primary tumor and intravasate into vasculature by undergoing an epithelial-to-mesenchymal transition (EMT), which triggers dissolution of epithelial polarity complexes, assembly of vimentin intermediate filaments, and establishment of a front-rear polarity axis defined by a peri-nuclear, compact Golgi organelle and leading and trailing edges enriched in focal adhesions (FAs) and other actin-based cytoskeletal structures that facilitate attachment to extra-cellular matrix proteins and promote cell motility[2–4]. Transitions between epithelial and mesenchymal states require precise spatial and temporal control of protein transport through endocytic recycling and retrograde vesicular trafficking pathways that facilitate plasma membrane (PM) dynamics, including the recy-cling of FAs from trailing to leading edges of a motile cell[5–7].

EMT is initiated by transcription factor families (e.g., ZEB, SNAIL, and TWIST) that silence the expression of epithelial polarity complexes (e.g., E-cadherin, Crumbs, and Claudins) and microRNAs (e.g., miR-200 family, miR-34a, miR-206, and miR-148a) that target stemness- and motility-inhibiting genes and EMT-activating transcription factors themselves, creating an adaptive, feed-forward regulatory system that controls reversible switching between epithelial and mesenchymal states[8–14]. How-ever, the way in which EMT-activating transcription factors govern protein transport through vesicular trafficking pathways to establish a front-rear polarity axis remains unclear.

Here, we postulated that EMT-activating transcription factors control endocytic vesicular trafficking networks to establish a front-rear polarity axis that facilitates motility. We tested this hypothesis in murine and human lung adenocarcinoma (LUAD) cell lines at distinct positions on the EMT spectrum; LUAD cells classified as "epithelial" have uniformly epithelial gene expression patterns and exhibit low metastatic propensities, whereas those classified as "mesenchymal" exhibit partial EMT features

characterized by bi-phenotypic gene expression patterns (e.g., high CDH1, CDH2, and VIM), a capacity to undergo EMT or the reverse process in response to extracellular cues, and an aggres-sive metastatic propensity driven by high levels of the EMT-activating transcription factor ZEB1[2,15–17]. We find that high ZEB1 levels accelerate post-endocytic vesicle trafficking to a perinuclear compartment where cargos are routed for recycling to the PM, degradation in lysosomes, or retrograde trafficking to the Golgi apparatus. We show that ZEB1 influences the intracellular fate of a receptor tyrosine kinase that undergoes endocytosis following ligand-binding and identify a ZEB1-dependent tran-scriptional program that accelerates vesicular trafficking through the endocytic recycling and retrograde pathways and thereby facilitates the establishment of a front-rear polarity axis. Thus, ZEB1 influences vesicular trafficking dynamics to execute cell polarity change.

## Results

**ZEB1 enhances endocytosis of PM-bound proteins.** We initially carried out a surface biotinylation assay on an epithelial LUAD cell line (393P) that undergoes EMT in response to ectopic ZEB1 expression[18]. Cells (393P_ZEB1 or 393P_vector) were treated on ice with a cleavable and membrane-impermeable biotin moiety. After transferring to 37 °C to initiate endocytosis, cells were washed to remove residual PM-bound biotinylated proteins, fixed, and stained with fluorescent streptavidin, which showed that ectopic ZEB1 expression accelerated the endocytosis and intracellular transport of biotinylated proteins to a peri-nuclear region that co-localized with the Golgi (Fig. 1a–c).

**ZEB1 accelerates vesicular trafficking through endocytic recy-cling and retrograde pathways.** In the surface biotinylation studies, endocytosed proteins were primarily perinuclear in 393P_ZEB1 cells and peripheral in 393P_vector cells (Fig. 1a–c and Supplementary Fig. 1a), which led us to speculate that ZEB1

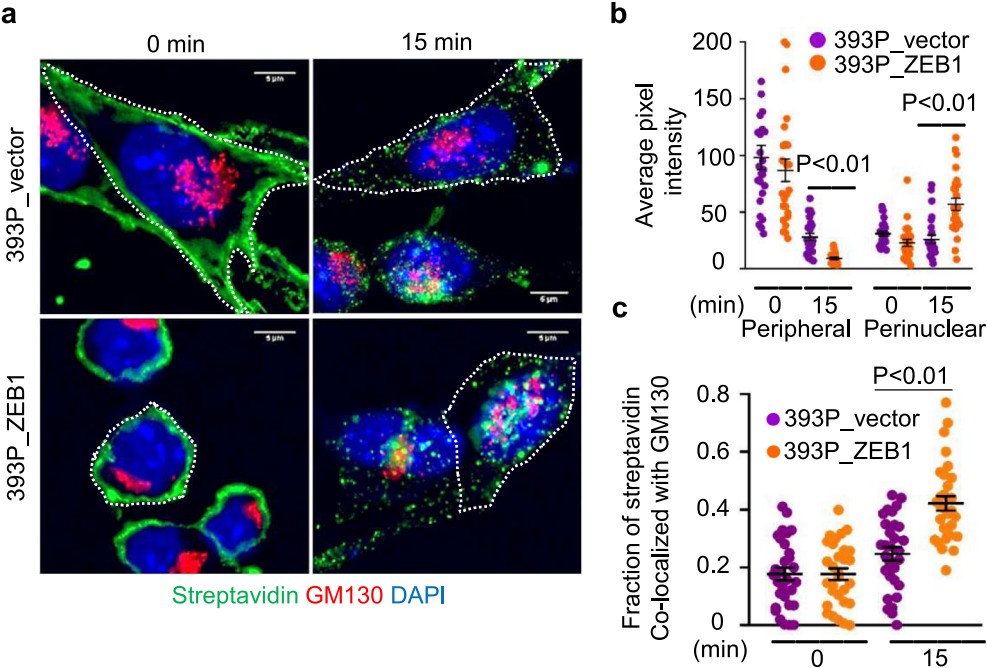

**Fig. 1 ZEB1 enhances the endocytosis of PM-bound proteins. a** Merged confocal micrographs taken before ($T = 0$) and 15 min after initiating endocytosis of labeled biotin. Cells treated with Alexa-488 labeled streptavidin and stained with anti-GM130 to detect endocytosed biotin and Golgi, respectively. Cells are outlined (dashed lines). Scale bars: 5 μm. **b** Biotin signal intensities in perinuclear and peripheral compartments in each cell (dot). Values represent maximal signal intensities in each cell. $n = 25$ cells from 3 independent experiments. **c** Fractions of total biotinylated proteins that colocalized with GM130 at each time point. $n = 30$ cells from 3 independent experiments. Data are presented as mean values ± SEM; $P$ values, two-tailed Student's $t$ test.

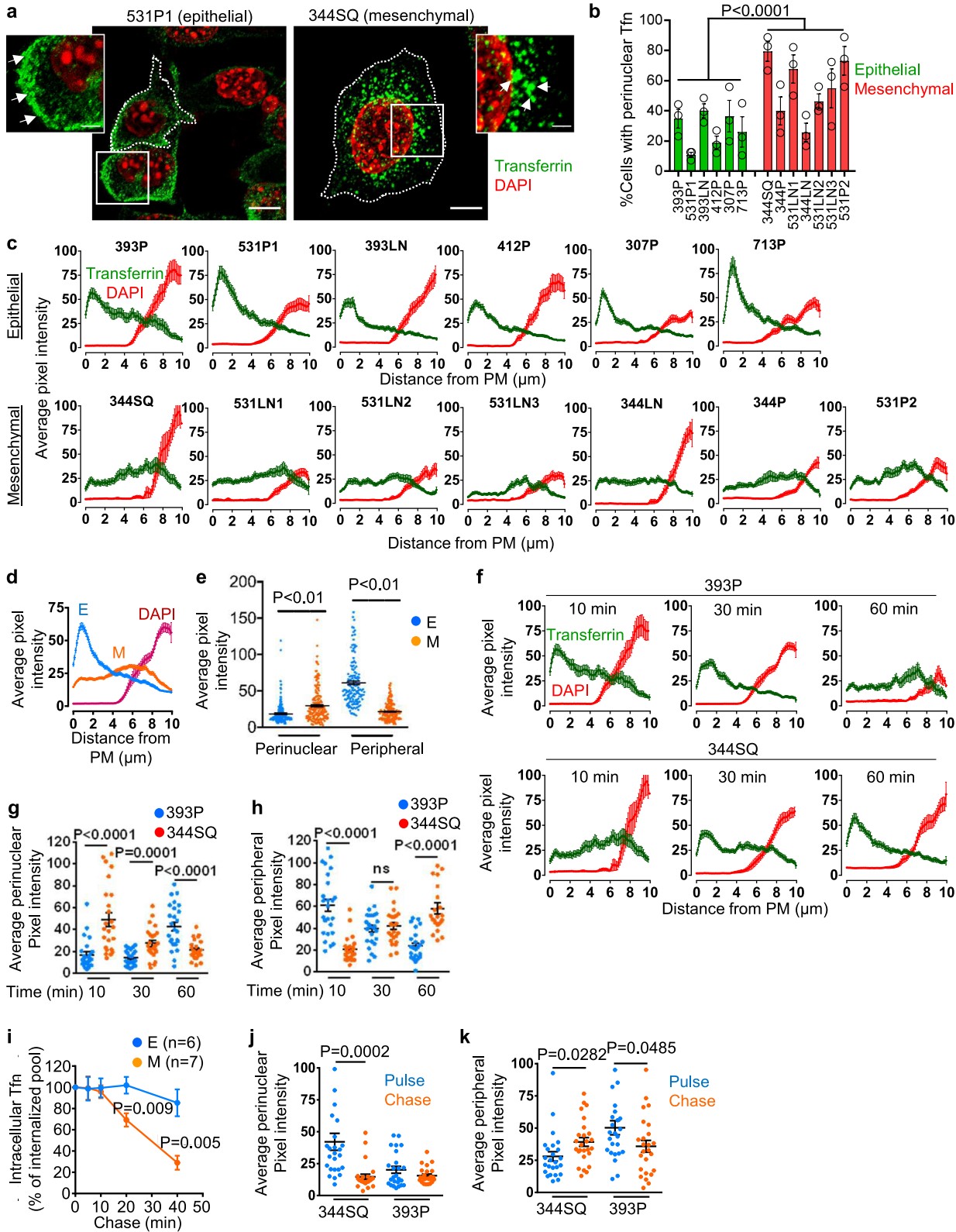

accelerates intracellular trafficking of endocytic vesicles. To address this possibility, we treated LUAD cell lines on ice with Alexa 568-labeled transferrin (Tfn), which binds to the transferrin receptor and is internalized and transported back to PM via the endocytic recycling pathway[19]. After transferring cells to 37 °C to induce endocytosis, we quantified Tfn endocytosis rates and found that they did not differ significantly between epithelial and

mesenchymal LUAD cells (Supplementary Fig. 1b), but Tfn localization at 10 min was peripheral in epithelial LUAD cells and peri-nuclear in mesenchymal LUAD cells (Fig. 2a–e and Supplementary Fig. 1c–e). To determine whether the distinct Tfn localization patterns resulted from differences in vesicular trafficking speeds, we quantified Tfn localization patterns 10, 30, and 60 min after initiation of endocytosis in mesenchymal 344SQ cells

**Fig. 2 Distinct transferrin (Tfn) trafficking patterns in epithelial and mesenchymal lung adenocarcinoma (LUAD) cells. a, b** Confocal micrographs taken 10 min after initiating endocytosis of Alexa 568-labeled Tfn. Insets illustrate peripheral and perinuclear Tfn in 531P1 cells and 344SQ cells, respectively. Scale bars: 10 μm, 3 μm (inset). Cells are outlined (dotted lines) (**a**). Each cell line is scored based on the percentage of total cells that have perinuclear Tfn. $n = 3$ independent experiments, 100 cells scored per experiment (**b**). **c** Alexa 568-labeled Tfn and DAPI signal intensities (y-axis) plotted on lines drawn from the plasma membrane (PM) inwards (x-axis) in cells fixed 10 min after initiating endocytosis. Results are averages of 3 linescans per cell, 19 cells per cell line. **d** Tfn and DAPI signals from (**c**) were expressed as mean values for epithelial ("E") and mesenchymal ("M") LUAD cell lines. **e** Tfn signal intensities in perinuclear and peripheral compartments of each cell (dot). Values represent the maximal signal intensities in each cell ($n = 152$ epithelial cells and 177 mesenchymal cells from 3 independent experiments). **f** Alexa 568-labeled Tfn and DAPI signal intensities determined as described in (**c**) 10, 30, and 60 min after initiating endocytosis. **g, h** Maximum signal intensities in peripheral (**g**) and perinuclear (**h**) compartments of each cell (dot). $n = 25$ cells from 3 independent experiments. **i** In-cell ELISA of intracellular Tfn levels. Results normalized based on total protein content. $n = 6$ (epithelial), or 7 (mesenchymal) cell lines. **j, k** Tfn staining intensities in perinuclear (**j**) and peripheral (**k**) compartments of each cell (dot). Cells were pulsed for 10 min with Alexa 568-labeled Tfn and chased for 10 min with unlabeled Tfn. Signal intensities after pulse alone (blue) or pulse/chase (red) were determined. $n = 25$ cells from 3 independent experiments. Data are presented as mean values ± SEM; P values, two-tailed Student's t test.

and found that the endocytosed Tfn reached the perinuclear region within 10 min and recycled back to the PM within 60 min, whereas Tfn trafficking was significantly slower in epithelial 393P cells, which required a full 60 min for the endocytosed Tfn to reach the perinuclear region (Fig. 2f–h). Findings from in-cell ELISA (Fig. 2i) and pulse-chase experiments (Fig. 2j, k) confirmed these results. Thus, endocytic recycling vesicular transport was faster in mesenchymal than epithelial LUAD cells.

Tfn receptor levels and Tfn-binding activities were similar in epithelial and mesenchymal LUAD cells (Supplementary Fig. 2a, b), arguing that the accelerated Tfn recycling did not result from enhanced Tfn-binding activity in mesenchymal cells. However, peripheral Tfn localization was correlated negatively with ZEB1 levels in the murine LUAD cell line panel (Fig. 3a), and Tfn labeling studies on epithelial (393P, 307P) and mesenchymal (344SQ) LUAD cell lines subjected to ZEB1 gain- and loss-of-function, respectively[2,18], showed that ZEB1 caused a perinuclear shift in Tfn localization (Fig. 3b–g). In pulse-chase studies on 344SQ cells, ZEB1 depletion resulted in a substantial delay in Tfn trafficking to the perinuclear region (Fig. 3h, i). Similar findings were observed in Madin–Darby canine kidney (MDCK) epithelial cells subjected to ectopic ZEB1 expression (Supplementary Fig. 2c, d). In live-cell imaging studies on 393P cells that express the mCherry-tagged early endosomal marker Rab5, ectopic ZEB1 expression increased Rab5⁺ vesicle speed and directionality (Fig. 3j, k and Supplementary Fig. 2e, Supplementary Movie 1). Thus, ZEB1 accelerates vesicular trafficking through the endocytic recycling pathway.

Endocytosed cargos can be either recycled to the PM or routed to lysosomes for degradation[20]. Lysosomal maturation is associated with a luminal acidification process that is critical for lysosomal functions[21]. To determine whether ZEB1 hastens the maturation of early endosomes into lysosomes, we utilized Tfn tagged with pHrodo Green, a fluorescent pH indicator that increases its intensity in vesicles with lower pH. We found that, in ZEB1 gain- and loss-of-function studies, ZEB1 increased pHrodo Green signal intensity (Fig. 4a–c and Supplementary Movie 2), which suggests that ZEB1 regulates late endosomal/lysosomal vesicle maturation.

Based on the above evidence that ZEB1 accelerates endocytosis of PM-bound proteins and hastens vesicular trafficking through the endocytic recycling and lysosomal pathways, we speculated that ZEB1 influences the intracellular fate of receptor tyrosine kinases that undergo endocytosis following ligand-binding and are either recycled back to the PM or delivered to lysosomes for degradation[6]. To address this possibility, we assessed the intracellular trafficking of MET, which is routed through endocytic recycling and lysosomal pathways with defined kinetics[22]. We carried out siRNA-mediated ZEB1 depletion studies on a mesenchymal human LUAD cell line (H1299),

treated ZEB1-deficient and -replete cells with hepatocyte growth factor (HGF) to initiate MET endocytosis, and imaged the endocytosed MET at multiple time points to quantify its co-localization with early endosomes (EEA1⁺), recycling endosomes (Rab11⁺), and late endosomes/lysosomes (LAMP1⁺). We found that ZEB1 deficiency delayed MET trafficking through the endocytic recycling compartment and prevented MET accumulation in lysosomes (Fig. 5a, b and Supplementary Fig. 3). In line with these findings, HGF-induced MET protein degradation was delayed in ZEB1-deficient cells (Fig. 5c). Thus, ZEB1 drives MET turnover by enhancing HGF-induced MET endocytosis and delivery of endocytosed MET to the lysosomal compartment.

Early endosomes that mature to late endosomes can route cargos to the Golgi and endoplasmic reticulum via the retrograde pathway[23–25]. To assess whether ectopic ZEB1 expression influences retrograde vesicular trafficking kinetics, we labeled LUAD cells with fluorescent cholera toxin B (CTxB), a retrograde cargo[26], and found that CTxB was transported to the Golgi more rapidly in 393P_ZEB1 cells than 393P_vector cells (Fig. 6) and in mesenchymal than epithelial human LUAD cell lines (Supplementary Fig. 4, Supplementary Movie 3), indicating that ZEB1 accelerates cargo trafficking through the retrograde vesicular pathway.

**ZEB1 relieves vesicular trafficking regulators from microRNA-dependent silencing to accelerate vesicular transport and establish a front-rear polarity axis.** Because ZEB1 silences diverse microRNAs that govern complex cellular processes[27,28], we assessed whether microRNAs serve as intermediates in a ZEB1-driven transcriptional program that governs vesicular trafficking. We found that Tfn localization in 344SQ cells shifted from the perinuclear region to the periphery following ectopic expression of miR-200 or miR-206 but not miR-148a (Fig. 7a–c), which are known transcriptional targets of ZEB1[2,16]. To identify microRNA targets that govern vesicular trafficking, we analyzed an RNA sequencing dataset (393P_ZEB1 versus 393P_vector, GSE102337) and identified known regulators of vesicular trafficking that are upregulated by ZEB1. These included genes that are predicted miR-200 targets (www.targetscan.org) and are components of an EMT-associated gene expression signature in the TCGA human LUAD cohort (Supplementary Fig. 5a). Quantitative PCR analysis carried out on 393P_ZEB1 cells and 393P_vector cells confirmed that 393P_ZEB1 cells have higher expression levels of multiple vesicular trafficking regulators (Fig. 7d), including KIF13A, a plus-end-directed microtubule-dependent kinesin motor protein that enhances intracellular vesicle transport from the perinuclear region to the PM[29–33], and AP1S2, a clathrin-associated adapter protein complex subunit that recognizes sorting signals within the cytosolic tails of transmembrane cargo molecules to control cargo transport

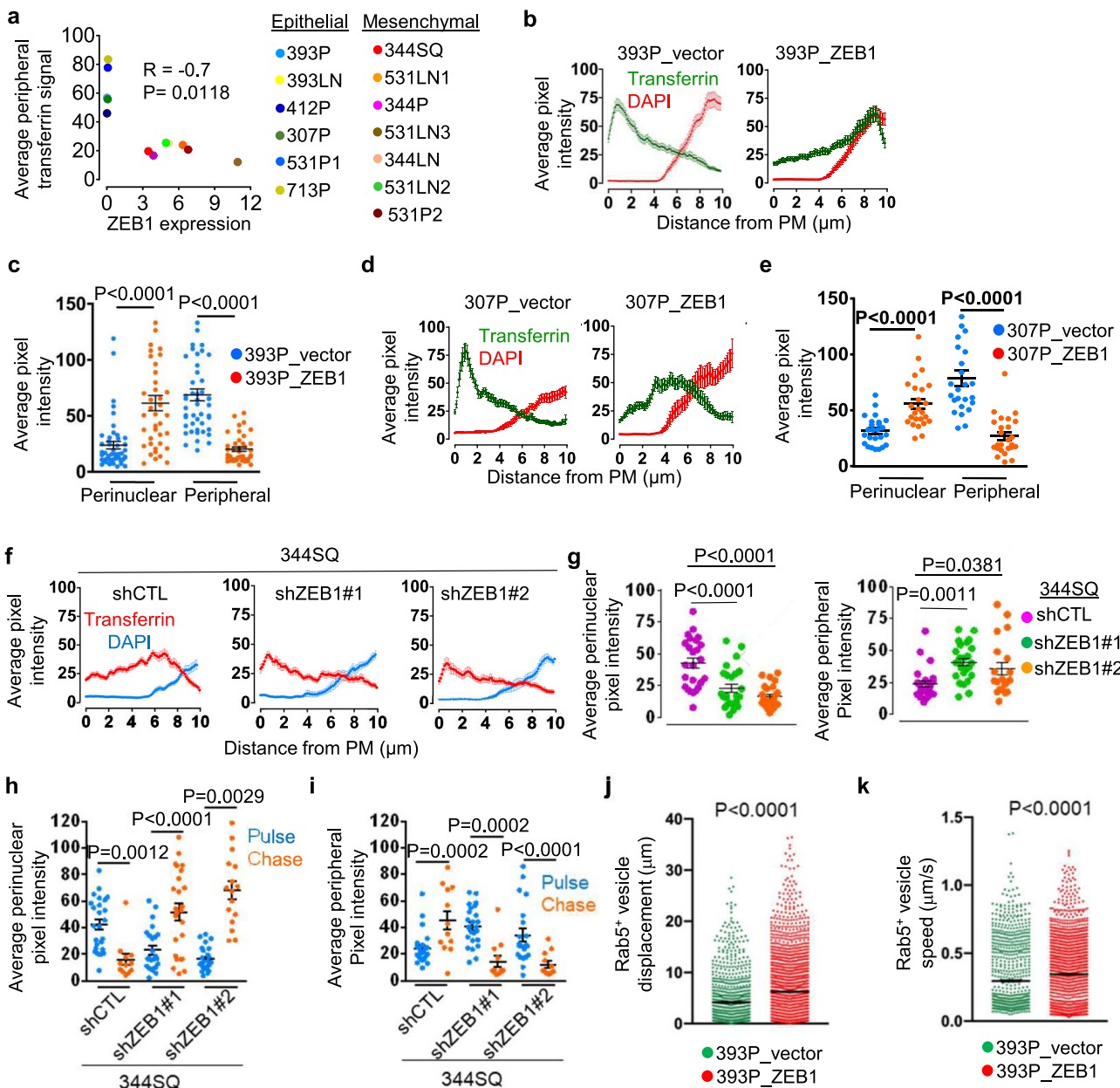

**Fig. 3 ZEB1 accelerates vesicular trafficking through the endocytic recycling pathway. a** Correlation of endogenous ZEB1 expression levels with Alexa 568-labeled Tfn signal intensities in peripheral compartments of murine LUAD cell lines fixed 10 min after initiating endocytosis. **b** Alexa 568-labeled Tfn and DAPI signal intensities (*y*-axis) on straight lines drawn from the PM inwards (*x*-axis) in 393P cells that have an ectopic expression of ZEB1 or empty vector. Cells fixed 10 min after initiating endocytosis. Results represent averages of 3 linescans per cell, ≥40 cells per cell line. **c** Tfn signal intensities in perinuclear and peripheral compartments of each cell (dot). Values represent the maximal signal intensities in each cell. *n* = 47 cells (393P_vector) or 40 cells (393P_ZEB1) cells from 3 independent experiments. **d, e** Alexa 568-labeled Tfn linescan plots (**d**) and perinuclear and peripheral signal intensities (**e**) in 307P cells that have an ectopic expression of ZEB1 or empty vector and were fixed 10 min after initiating endocytosis. *n* = 25 cells from 3 independent experiments. **f, g** Alexa 568-labeled Tfn linescan plots (**f**) and perinuclear and peripheral signal intensities (**g**) in ZEB1 (shZEB1 #1 or #2)- or control (shCTL) shRNA-transfected 344SQ cells fixed 10 min after initiating endocytosis. *n* = 25 cells (shZEB1 #1) or 20 cells (shZEB1 #2) from 3 independent experiments. **h, i** Tfn signal intensity in shZEB1- and shCTL-transfected 344SQ cells pulsed for 10 min with Alexa 568-labeled Tfn and chased for 10 min with unlabeled Tfn. Signal intensities in perinuclear (**h**) and peripheral (**i**) compartments after pulse alone (blue) or pulse/chase (red) were determined. Values represent the maximal signal intensities in each cell. For pulse/chase (**h**), *n* = 12 cells (shCTL), 25 cells (shZEB1 #1), or 16 cells (shZEB1 #2) from 2 independent experiments. For pulse/chase (**i**), *n* = 12 cells from 2 independent experiments. **j, k** Displacement (**j**) and speed (**k**) of each Rab5+ vesicle (dot) in 39P_vector cells and 393P_ZEB1 cells that express mCherry-tagged Rab5, an early endosomal marker. Data represent mean ± S.E.M (*n* = 3 movies, 2 cells per movie, 600 frames per cell). Data are presented as mean values ± SEM; *R* value, Spearman; *P* values, two-tailed Student's *t* test; or ANOVA (**g**).

between endosomes and the *trans*-Golgi network[34–36]. In line with evidence that the KIF13A and AP1S2 3′-untranslated regions have functional miR-200 binding sites[37], ectopic miR-200 expression mitigated ZEB1-induced KIF13A and AP1S2

expression (Fig. 7e). To assess KIF13A and AP1S2 as mediators of ZEB1-driven vesicular trafficking, we subjected 393P_ZEB1 cells to siRNA-mediated depletion of KIF13A or AP1S2 (Supplementary Fig. 5b) and found that KIF13A depletion abrogated

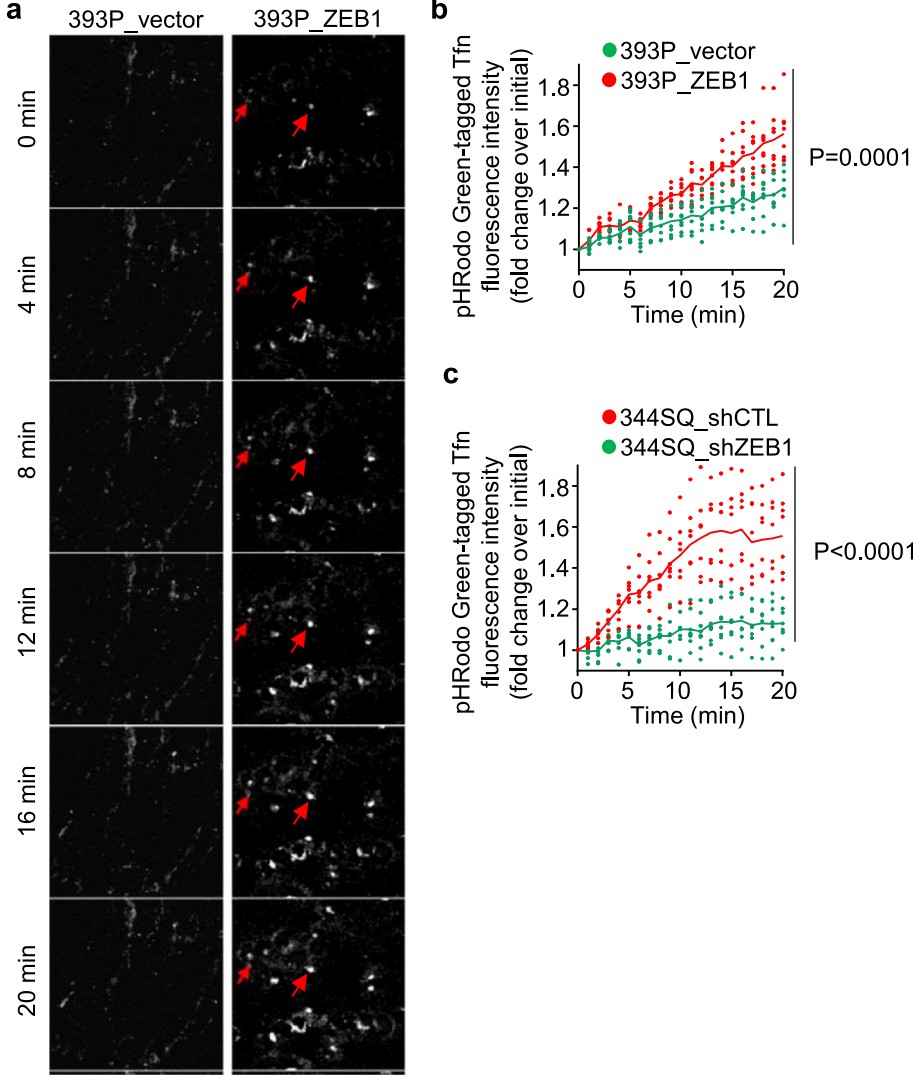

**Fig. 4 ZEB1 accelerates endosomal acidification. a** Montage of live-cell confocal microscopic images of 393P cells that have ectopic expression of ZEB1 or empty vector and were treated with pHRodo Green-tagged Tfn. Gains in signal intensity over time (red arrows) mark vesicles that have reduced intravesicular pH and enhanced maturation. Scale bars, 10 μm. **b, c** Quantification of pHRodo Green-tagged Tfn signal intensities following ZEB1 gain-of-function (**b**) or ZEB1 loss-of-function (**c**). Signal intensities in each cell (dot) determined at each time point after pHRodo Green-tagged Tfn treatment initiation. n = 8 cells from 4 independent experiments. P values, two-way ANOVA.

ZEB1-driven acceleration of Tfn recycling to the PM but not CTxB retrograde trafficking to the Golgi, whereas AP1S2 depletion mitigated ZEB1-driven retrograde trafficking of CTxB to the Golgi but not Tfn recycling to the PM (Fig. 7f–j and Supplementary Fig. 5c, d). Thus, ZEB1 accelerates trafficking through endocytic recycling and retrograde pathways by relieving KIF13A and AP1S2 from microRNA-mediated silencing.

In the LUAD cell line panel, a peripheral Tfn staining pattern was correlated with low metastatic activity and the absence of compact Golgi organelles that are essential for ZEB1-driven cell motility (Fig. 8a, b)[2], which led us to assess whether ZEB1-dependent vesicular trafficking networks influence the establishment of a front-rear polarity axis. By utilizing scratch-wound assays in which cells on the wound front orient their Golgi and FAs toward the leading edge[38–40], we found that siRNA-mediated depletion of KIF13A or integrin α5(ITGα5), a known driver of cell polarity[41–43], mitigated FA polarization toward the leading edge of 344SQ cells (Fig. 8c, d and Supplementary Fig. 6a–c). Based on evidence that nascent FAs on the wound front undergo a maturation process marked by recruitment of a protein complex containing, among other components, integrins, and a surrounding actin mesh[44,45], we quantified nascent FAs on the wound front and found that depletion of KIF13A or ITGα5 in 344SQ cells increased FA numbers and reduced FA sizes (Fig. 8e–g), suggesting a loss of FA maturation. Lastly, depletion of KIF13A, AP1S2, or ITGα5 reduced 393P_ZEB1 cell motility speed and directional persistence on fibronectin-coated plates, inhibited 344SQ cell motility and invasiveness in Boyden chambers, and suppressed 393P_ZEB1 cell invasiveness in 3-dimensional collagen (Fig. 8h–l and Supplementary Fig. 6d, e). These effects were recapitulated by treatment with ikarugamycin (Supplementary Fig. 6f, g), an inhibitor of clathrin-mediated endocytosis[46]. Thus, ZEB1-dependent vesicular trafficking drives the establishment of a front-rear polarity axis.

## Discussion

EMT is a transcriptionally governed process that initiates cell motility by generating a leading edge with actin-based cytoskeletal projections that cycle from back to front in motile cells[8,47].

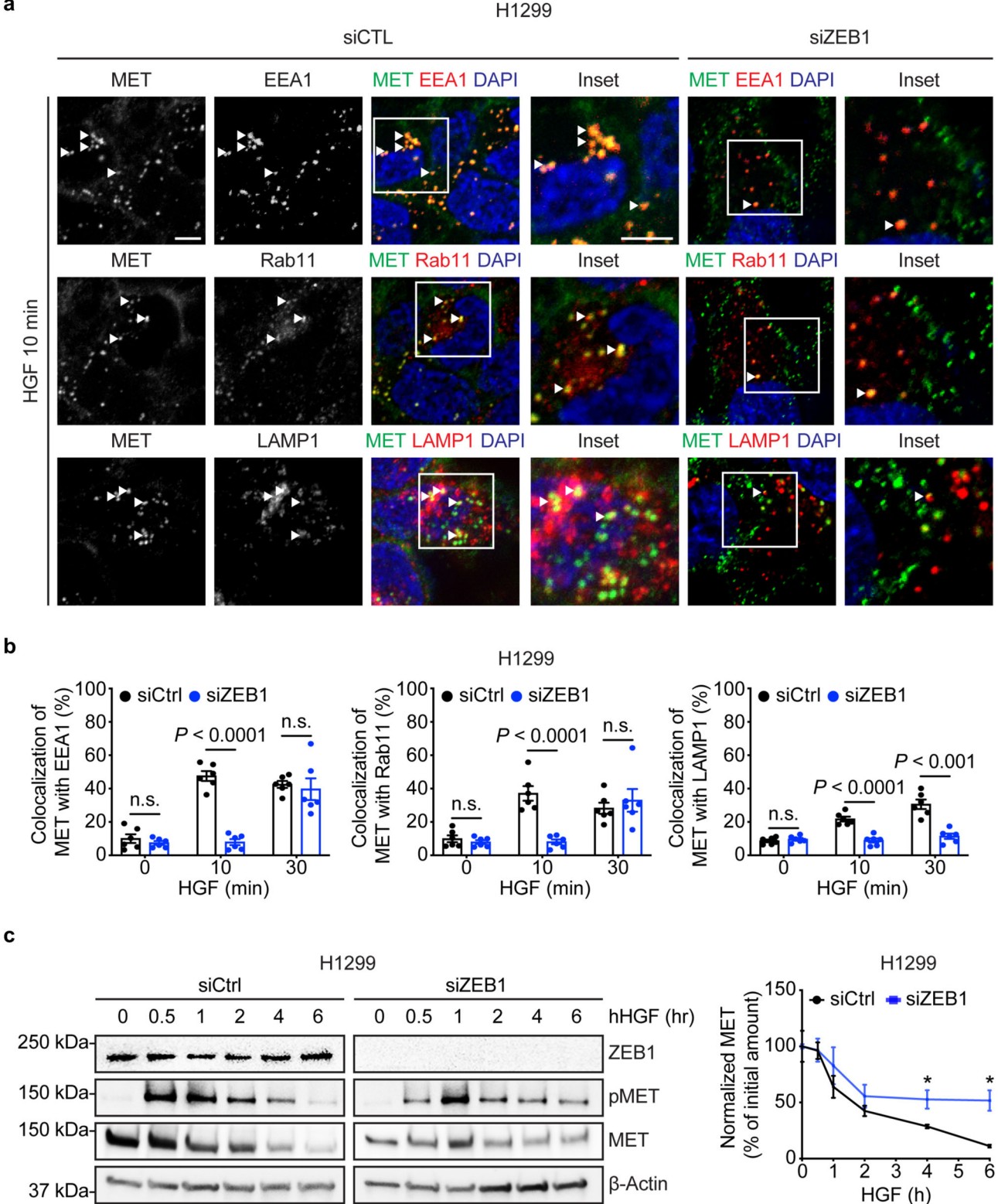

Such dynamics require precise spatial and temporal control of vesicular trafficking. Yet, the way in which EMT-activating transcription factors govern vesicular trafficking networks that affect cell polarity change remains unclear. Here, we show that the EMT activator ZEB1 is a key regulator of endocytic vesicular transport dynamics that establish a front-rear polarity axis and drive cell motility (Fig. 8m). In the context of our prior report[2],

these findings suggest that ZEB1 coordinates Golgi dynamics with accelerated cargo transport to execute cell polarity change.

The speed with which PM-bound proteins are endocytosed and recycled back to the PM is determined by minus- and plus- end-directed molecular motors that govern directional endosome movement along microtubules to and from the perinuclear compartment[6]. In concert with components of the clathrin

**Fig. 5 ZEB1 influences intracellular trafficking of MET. a** Confocal images of control (siCTL) and ZEB1 siRNA-transfected H1299 cells co-stained with antibodies against endogenous MET, EEA1, Rab11, and LAMP1. Cells were incubated at 4 °C with 250 ng/ml HGF for 30 min, washed, incubated at 37 °C for 10 min, fixed, and stained. Single-channel and merged images (left-to-right, panels 1–2 and 3–6, respectively). Boxed areas are magnified (insets). Scale bars, 5 μm. Co-localized structures in siCTL-transfected cells are indicated (arrowheads). **b** Quantification of **a** MET colocalization with EEA1 (left bar graph), Rab11 (middle bar graph), or LAMP1 (right bar graph). Values represent the percentage of total MET that colocalizes with each protein in each microscopic field (dot) of siCTL transfectants ($n = 26$ cells at 0 min, 52 cells at 10 min, and 60 cells at 30 min) and siZEB1 transfectants ($n = 24$ cells at 0 min, 28 cells at 10 min, and 29 cells at 30 min). Data represent mean ± SEM. **c** Western blot analysis of siCTL- and siZEB1-transfected H1299 cells treated with HGF (100 ng/ml) in the presence of cycloheximide (40 μg/ml) and incubated at 37 °C for the indicated times. Densitometric values of MET band intensities normalized to β-actin and expressed as a percentage of the initial amount ($T = 0$) (line plot). Data represent mean ± SEM ($n = 3$). P values, two-tailed Student's $t$ tests. *$P < 0.05$.

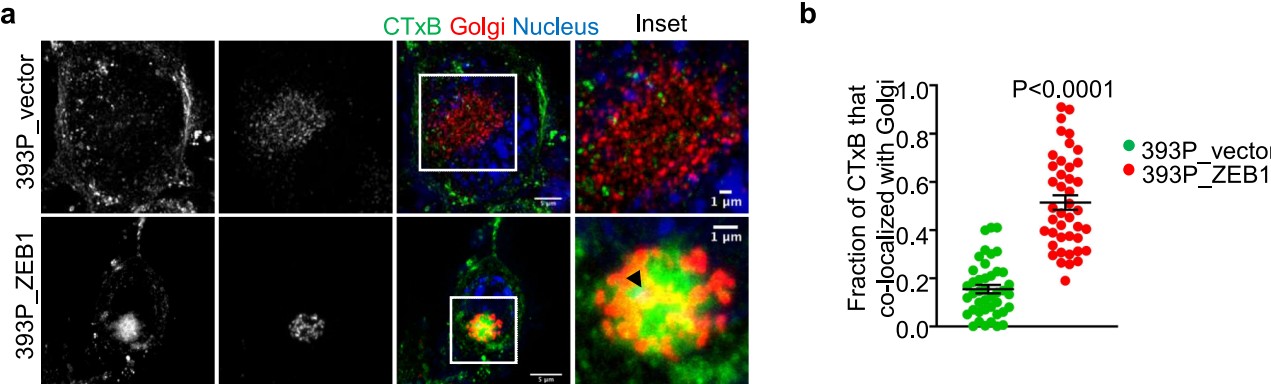

**Fig. 6 ZEB1 accelerates retrograde vesicular trafficking. a** Single-channel and merged confocal micrographs taken 1 h after initiating Alexa 488-labeled cholera toxin B (CTxB) treatment. Boxed areas are magnified (insets) to show greater CTxB co-localization with Golgi in 393P_ZEB1 cells than 393P_vector cells. Scale bars: 5 μm, 1 μm (inset). **b** Fraction of total CTxB that co-localizes with GM130, a Golgi marker, in each cell (dot). $n = 42$ cells from 3 independent experiments. Data are presented as mean values ± SEM; P values, two-tailed Student's $t$ test.

adapter AP-1 complex and the perinuclear recycling-associated Rabs, the kinesin motor proteins KIF13A and KIF13b regulate cargo trafficking from the perinuclear compartment towards the PM[29–33]. Findings presented here suggest that ZEB1 accelerates endocytosis of PM-bound proteins and drives endocytic recycling and retrograde trafficking of endocytosed proteins by relieving KIF13A and AP1S2 from microRNA-dependent silencing.

Endocytic vesicular trafficking plays a key role in establishing a front-rear polarity axis and determines the speed and persistence of directional cell migration[48–50]. Integrins are redistributed from the cell's back to front via an endocytic pathway that is regulated by Numb, aPKC[51], and the Scribble and Exocyst protein complexes, which tether transport vesicles to sites of fusion with the PM[52–54]. Integrin dimeric complexes are transported in ways that facilitate FA recycling and cell motility[48]. For example, non-ligand-bound ITG-β1 remains anchored to substratum as the cell moves forward and is transported in a retrograde fashion from trailing to leading-edge, whereas ligand-bound ITG-β1 and ITGα5 are transported via the endocytic recycling pathway[55,56]. Findings presented here suggest that both vesicular trafficking pathways are governed by ZEB1-driven transcriptional programs that affect cell polarity change.

EMT-related gene expression signatures mark LUAD patients who have advanced disease and a worse clinical outcome[57]. Findings presented here suggest that EMT drives a vesicular trafficking program that contributes to poor prognostic features and might be targeted with selective antagonists of kinesin family members, integrin heterodimeric complexes, and FA kinases that are under clinical development[58–61].

## Methods
**Reagents.** Murine LUAD cell lines were derived previously[17]. We purchased MDCK cells and human LUAD cell lines from ATCC; conjugated and unconjugated transferrin, EZ-Link Sulfo-NHS-SS-Biotin, conjugated Streptavidin,

Lipofectamine 3000, fetal bovine serum (FBS), donkey serum, HEPES-buffered media, Live-cell imaging solution, Dulbecco's minimal essential medium (DMEM), phosphate-buffered saline (PBS), RPMI-1640, phalloidin, CtxB, Cell-Light reagents, type I collagen, SYBR Green, TRIzol, paraformaldehyde, D-Glucose, bovine serum albumin (BSA), DAPI, Hoechst, DRAQ5, and Triton X-100 from Thermo Fisher Scientific; puromycin from InvivoGene; Transwell and Matrigel-coated Boyden chambers from BD Biosciences; Glass-bottom dishes and multiwell plates from MatTek; Trypsin, G418, and Penicillin–Streptomycin solutions from Corning; qScript cDNA superMix from Quanta Biosciences; saponin and siRNAs from Sigma; 10 × Cell lysis buffer and protease/phosphatase inhibitor cocktail from Cell Signaling Technologies; Zenon Mouse IgG2a Labeling Kit from Thermo Fisher Scientific; primary antibodies against LAMP1 (#9091, 1:100 dilution for immunofluorescence), MET (#8198, 1:1000 dilution for WB), phospho-MET (#3077, 1:1000 dilution for WB), phospho-Paxillin (Tyr118) (#2541, 1:100 dilution for immunofluorescence), and Rab11 (#5589, 1:100 dilution for immunofluorescence) from Cell Signaling Technologies; and primary antibody against MET (#AF276, 1:100 dilution for immunofluorescence) and recombinant human HGF protein (#294-HG-005/CF) from R&D Systems. Alexa Fluor-tagged secondary antibodies (#A-11055, 1:1000 dilution for immunofluorescence; #A-32787, 1:1000 dilution for immunofluorescence; #A-10042, 1:500 dilution for immunofluorescence) from Thermo Fisher Scientific. Horseradish peroxidase (HRP)-conjugated secondary antibodies (#7074P2, 1:1000 dilution for WB; #7076P2, 1:1000 dilution for WB) from Cell Signaling Technologies. mCh-Rab5 was a gift from Gia Voeltz (Addgene plasmid #49201; http://n2t.net/addgene:49201; RRID:Addgene_49201)[62].

**Cell culture.** Murine and human LUAD cell lines were cultured in RPMI-1640 with 10% FBS. MDCK cells were cultured in DMEM with 10% FBS. Cells were maintained at 37 °C in an incubator with a humidified atmosphere containing 5% $CO_2$. Cells were transfected with vectors using Lipofectamine 3000, and stable transfectants were selected for 2 weeks using puromycin or G418. Cell-Light BacMam baculoviral Golgi-RFP was used at 30–50 particles per cell and incubated for 36–48 h before imaging.

**Quantitative RT-PCR.** Total RNA was isolated from cells using TRIzol and subjected to reverse transcription and qPCR analysis as described[2,16]. mRNA levels were normalized based on ribosomal protein L32 (Rpl32) mRNA. MicroRNA levels were quantified using stem–loop RT-PCR assays as described[2,16]. Primer sequences are listed in Supplementary Table 1.

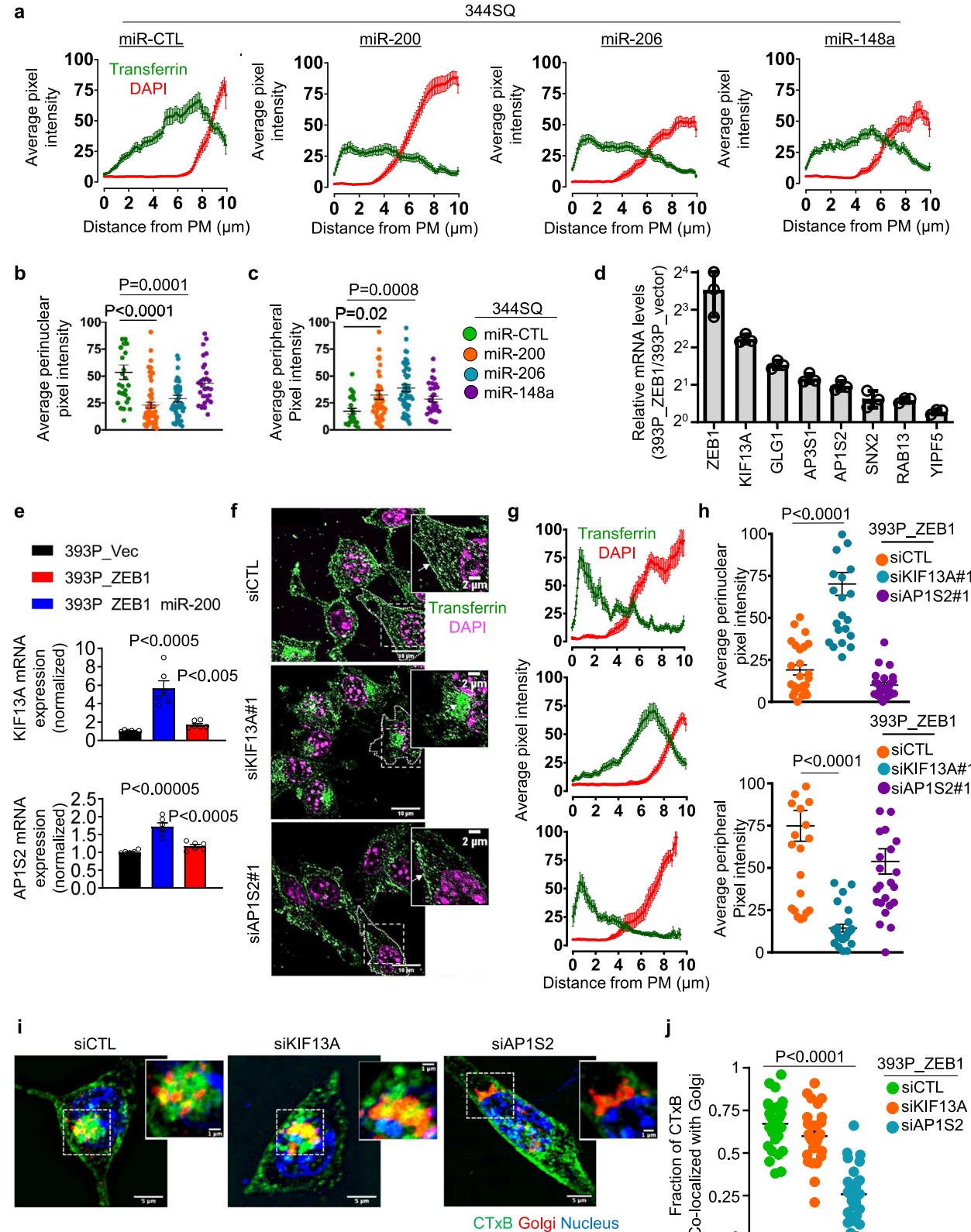

**Cell migration and invasion assays in Boyden chambers**. As described previously[2], migration and invasion assays were performed in Transwell and Matrigel-coated Boyden chambers, respectively. For single-cell tracing assays, $1 \times 10^4$ cells per well were seeded on fibronectin-coated, glass-bottom 24-well plates. Cells were imaged at 5-min intervals for 24 h in DIC mode using a $20 \times 0.75$ NA Air objective on an Eclipse TiE inverted microscope (Nikon) using NIS Elements V4.40 software. For Boyden chambers, $2 \times 10^4$ cells were seeded in the upper wells of Transwell and Matrigel chambers, respectively (BD Biosciences) and allowed to migrate toward 10% FBS in the bottom wells. After 8–10 h of incubation, migrating, or invading cells were stained with 0.1% crystal violet, photographed, and counted.

**Multicellular aggregate invasion assays in three-dimensional collagen**. Multicellular aggregates were generated in hanging drops by creating suspensions of 2000 cells in 25-μl droplets for 48 h to ensure multicellular aggregation, at which

**Fig. 7 ZEB1 relieves regulators of vesicular trafficking from microRNA-dependent silencing. a** Alexa 568-labeled Tfn and DAPI signal intensities (*y*-axis) on lines drawn from the PM inwards (*x*-axis) in 344SQ cells that have an ectopic expression of microRNAs or non-coding control oligomer (miR-CTL). Cells fixed 10 min after initiating endocytosis. Results represent averages of three linescans per cell. **b, c** Dot plots illustrate Tfn signal intensity in perinuclear (**b**) and peripheral (**c**) compartments in each cell (dot). *n* = 26 cells (miR-CTL), 53 cells (miR-200), 50 cells (miR-206), or 29 cells (miR-148a) from 3 independent experiments (**a–c**). **d** qPCR analysis of mRNA levels of vesicular trafficking regulators in 393P cells that have an ectopic expression of ZEB1 or empty vector. *n* = 3 independent experiments. **e** qPCR analysis of KIF13A and AP1S2 mRNA levels in 393P_ZEB1 cells that have an ectopic expression of the miR-200b/a/429 cluster or empty vector (miR-CTL). *n* = 6 independent experiments. **f, g** Merged confocal micrographs taken 1 h after initiating endocytosis. Scale bars: 10 µm, 2 µm (inset). Alexa 568-labeled Tfn has recycled back to the PM in 393P_ZEB1 cells transfected with siCTL or siAP1S2 but not siKIF13A, which induced a perinuclear staining pattern, suggesting a substantial delay in Tfn recycling, a conclusion supported by Alexa 568-labeled Tfn and DAPI linescan plots (**g**). *n* = 25 cells from 3 independent experiments. **h** Tfn signal intensities in perinuclear (top plot) and peripheral (bottom plot) compartments of each cell (dot). Values represent the maximal signal intensities in each cell. *n* = 25 cells from 3 independent experiments. **i** Merged confocal micrographs taken 1 h after initiating Alexa 488-labeled cholera toxin B (CTxB) treatment. Boxed areas are magnified (insets) to show that CTxB co-localization with Golgi is abrogated by siAP1S2 but not siKIF13A. Scale bars: 5 µm, 1 µm (inset). **j** Fractions of total CTxB that co-localized with GM130 in each cell (dot). *n* = 25 cells from 3 independent experiments. Data are presented as mean values ± SEM, or mean values ± SD (**d**); *P* values, one-way ANOVA.

point the aggregates were washed with medium, mixed with 2 mg/ml of rat-tail collagen I solution (BD Biosciences), and polymerized at 37 °C in an incubator. Complete medium was added, and the aggregates were imaged under a bright-field microscope 12 h after seeding. Multi-cellular protrusions and single migratory cells were manually counted.

**In-cell ELISA.** These assays are based on a previous report[63]. To quantify Tfn endocytosis, $2 \times 10^4$ cells per well were seeded on gelatin-coated 96-well plates and incubated for 15 h before transferring to ice and treating with 5 µg/ml biotinylated-Tfn in live-cell imaging media (phenol-red free HEPES-buffered media containing 140 mM NaCl, 2.5 mM KCl, 1.8 mM CaCl2, 1 mM MgCl2, 20 mM HEPES, mOsm 300, pH 7.4 and supplemented with 0.2% BSA and 5 mM D-Glucose) at 37 °C for various time periods. Cells were transferred to the ice to stop endocytosis, and surface-bound Tfn was removed by three washes with stripping buffer (0.2 M acetic acid, 0.2 M NaCl, pH 2.5). Cells were then washed with ice-cold PBS, fixed in 4% paraformaldehyde for 20 min, and permeabilized with 0.1% Triton X-100 for 10 min. Microplates were analyzed in Molecular Devices SpectraMax M series multi-mode microplate reader using SoftMax Pro software. Endocytosed Tfn was assessed using Streptavidin-HRP and quantified as the percentage of total surface-bound Tfn measured in parallel. To correct for well-to-well variability in cell number, results were normalized based on total protein levels with a BCA readout at 560 nm. To quantify Tfn recycling, cells were washed with ice-cold PBS on ice after the stripping step described above and incubated with 2 mg/ml holo-Tfn at 37 °C for various time periods. Cells were washed in stripping buffer and PBS, fixed, and detected as described above. Intracellular biotinylated-Tfn levels were expressed relative to total internalized Tfn. Recycled Tfn is defined as the fraction of total internalized Tfn that is lost after the final stripping.

**Vesicular trafficking assays.** For imaging-based assays of Tfn endocytic trafficking and pulse-chase recycling, $1 \times 10^5$ cells per well were seeded on type I collagen-coated glass-bottom dishes and incubated for 12–18 h. Biotinylated Tfn was replaced with 25 µg/ml of Alexa Fluor 568-labeled Tfn[64]. Cells were permeabilized, labeled with DAPI, and imaged. For retrograde vesicular trafficking assays, cells were placed on ice for 10 min, incubated with 1ug/ml fluorescent CTxB for 10 min, washed in ice-cold live-cell imaging medium, and subjected to live-cell imaging immediately or incubated at 37 °C for 1 h and fixed on ice.

**MET trafficking assay.** Adapted from a previous report[65], the cells were washed three times with PBS and then starved in RPMI 1640 without FBS for 16 h at 37 °C. Cells were incubated with 250 ng/ml HGF in RPMI 1640 medium for 30 min at 4 °C, washed with cold PBS, and then incubated in pre-warmed RPMI 1640 medium at 37 °C for the indicated times. Cells were then washed with ice-cold PBS to stop the chase, formalin-fixed, stained with antibodies against MET, EEA1, Rab11, and LAMP1, and imaged.

**Western blotting.** Cells cultured in each well of a 6-well plate at 80% confluency were washed three times with PBS and harvested/resuspended in 150–200 µl of 2 × Laemmli buffer (Bio-Rad). The cell lysate was boiled for 10 min and loaded onto an SDS gel. After transferring to a nitrocellulose membrane (Bio-Rad), membranes were blocked with 5% milk in TBST buffer and were probed with primary antibodies diluted in 5% BSA in TBST buffer. HRP-conjugated secondary antibodies were used according to the manufacturers' instructions. Quantitative analysis was performed by using ImageJ software (NIH).

For HGF-induced degradation of MET, after siRNA transfection, the cells were seeded in each well of a 6-well plate containing RPMI 1640 with 10% FBS. Eight hours after seeding, cells were washed three times with PBS and starved in RPMI

1640 without FBS for 16 h. The cells then were untreated or treated with 100 ng/ml of HGF in the presence of cycloheximide (40 µg/ml) for the indicated times. After HGF stimulation, cells were washed three times with PBS and harvested/resuspended in 150–200 µl of 2 × Laemmli buffer. Cell lysates were subjected to Western blotting and image analysis as described above.

**Live and fixed cell imaging.** Cells were seeded on fibronectin- or type I collagen-coated cover glass (#1.5). Live cells were imaged in live-cell imaging media within incubation chambers. For fixed-cell imaging, cells were fixed using 4% paraformaldehyde for 10 min, permeabilized using 0.1% Triton X-100 for 5 min or with 0.1% saponin for the entire period, and blocked with 3% BSA/2% donkey serum, with or without saponin for 30 min. F-actin was detected using Alexa Fluor–conjugated phalloidin. Primary antibody incubation was performed in blocking buffer for 1 h at room temperature or overnight at 4 °C, followed by Alexa Fluor-conjugated secondary antibodies (1:500) in blocking buffer for 1 h at room temperature. Nuclei were counterstained with DAPI and a cover glass was mounted using Vectashield (H-1000). Each step was followed by washing three times with PBS, which was also used as the solvent in all steps.

**Microscopy.** Confocal imaging was performed on an A1 + platform (Nikon Instruments) equipped with 63 ×/1.4 NA Oil, 100 ×/1.45 NA Oil, and 20×/0.75 NA Air objectives; 405/488/561/640 nm laser lines; GaAsP detectors; and Okolab stage top incubator. Images were acquired using NIS-Elements software (Nikon instruments). Widefield and live-cell imaging were performed on an automated Eclipse TiE (Nikon Instruments) microscope equipped with the same objectives as above, LED light for illumination, DSQi2 CMOS camera for detection, and an Okolab cage incubation system. For some live-cell imaging experiments, an Andor Revolution XDi microscope (Oxford Instruments) equipped with Yokogawa WD spinning disc, Borealis illumination, 60 ×/1.35 NA Silicone objective; 405/488/561/640 nm laser lines; iXon Ultra 888 EMCCD and Zyla 4.2 sCMOS cameras; Andor iQ software; and stage top Tokai Hit incubator was used. Wherever applicable, Nyquist sampling criteria were followed to ensure proper sampling for deconvolution purposes. Routine widefield fluorescence, bright field, and phase-contrast imaging were performed on an IX71 microscope (Olympus).

**Image processing and quantitative analysis.** Raw images were iteratively deconvolved (Huygens Professional, Scientific Volume Imaging, VB) and further processed/analyzed in Imaris 9.6 (Bitplane software, Oxford instruments) with MATLAB XTensions and Fiji/ImageJ (www.imagej.nih.gov). Linescan plots[66,67] were generated by drawing multiple fixed-length and -width lines from the periphery to the center of cells in RGB overlay images using the RGB profile plot function in ImageJ. Cell outlines visible in inverted images of the Tfn channel (Supplementary Fig. 1F) were used to draw line profiles. Co-localization analysis (Huygens Professional) was plotted using Manders' coefficients expressed as the fraction of red or green pixels overlapping with each other. A particle analyzer (ImageJ) was used to measure FA parameters as described[68,69]. Time-lapse videos were analyzed for single-cell and vesicle trajectories in the Imaris spots tracking module in Surpass mode[70–73]. Golgi and FA polarity[39,74] were assessed in GM130- and phospho-paxillin-stained cells at the leading edge of scratch wounds by drawing lines that radiate from the center of the nucleus to either edge of the cell's migratory front. A cell was scored as polarized if the region between the two lines encompasses the majority of the GM130- or phospho-paxillin-stained structures. Co-localization of MET with endosomal markers was assessed in Imaris 9.6 (co-localized spots feature) to quantify pixels in different channels that overlap.

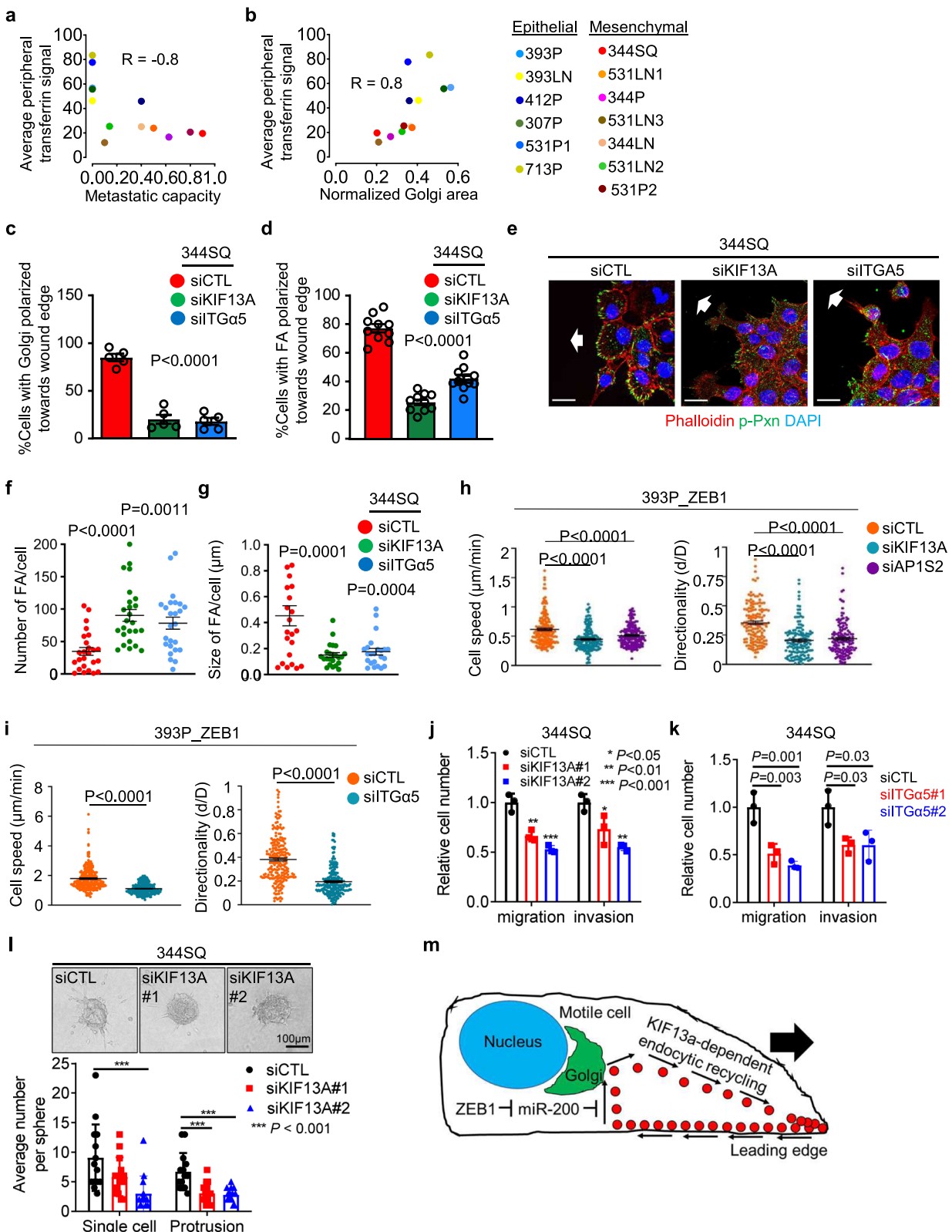

**Biotinylation.** Cell-surface biotinylation was performed as described[75]. 393P_vector cells and 393P_ZEB1 cells were treated for 30 min on ice with a cleavable and membrane-impermeable biotin moiety (Thermo Fisher Scientific EZ-Link™ Sulfo-NHS-SS-Biotin). Cells were either transferred to 37 °C to initiate endocytosis or left on ice as a control for total PM-associated biotinylated proteins ($T = 0$). After a stripping step to remove unincorporated biotin from the PM, cells were stained with fluorescent streptavidin and imaged.

**Statistics and reproducibility.** Unless mentioned otherwise, all micrographs, blots, and results shown are representative of at least three independently replicated experiments. Values shown are the means ± standard error of the means from triplicate samples or randomly chosen cells within a field unless stated otherwise. Statistical evaluations were carried out with Prism 8 (GraphPad Software, Inc.). Heat maps were generated using JavaTreeView[76]. Unpaired two-tailed Student $t$ tests or ANOVA were used to compare means for two or more groups, respectively, and $P$ values < 0.05 were considered statistically significant.

**Fig. 8 ZEB1-dependent vesicular trafficking facilitates the establishment of a front-rear polarity axis. a, b** Correlation of peripheral Tfn signal intensity with metastatic capacity (**a**) and Golgi area (**b**) in murine LUAD cell lines fixed 10 min after initiating endocytosis. **c** Percentages of cells on the wound front with Golgi organelles polarized toward the leading edge. $n = 5$ independent experiments. **d** Percentages of cells on the wound front with FAs polarized toward the leading edge. $n = 10$ independent experiments. **e** Merged confocal micrographs of siRNA-transfected 344SQ cells stained with anti-phospho-paxillin (p-Pxn) antibody to detect FAs. The direction of cell migration on advancing front (arrow). Scale bars: 20 μm. **f, g** Plot showing average FA number (**f**) ($n = 25$ cells from 3 independent experiments) and size **g** ($n = 23$ cells from 3 independent experiments) per cell in the siRNA-transfected 344SQ cells (**e**). **h, i** Cell speed and directionality for each cell (dot) measured from automated tracking of 393P_ZEB1 cells seeded at low density on the fibronectin-coated surface and transfected with siRNAs against KIF13A (**h**), AP1S2 (**h**), or ITGα5 (**i**). **h** $n = 180$ cells (speed) or 143 cells (directionality) from 3 independent experiments. **i** $n = 250$ cells (speed) or 225 cells (directionality) from 3 independent experiments. **j, k** Migration and invasion assays in Boyden chambers for 344SQ cells transfected with KIF13A (**j**) or ITGα5 (**k**) siRNAs. $n = 3$ independent experiments. **l** Invasive activities of siRNA-transfected 393P_ZEB1 cells seeded as multicellular aggregates in collagen gels. Invasions scored as single-cell or collectively invasive protrusions. $n = 14$ aggregates from 3 independent experiments. Scale bars: 100 μm. **m** Model illustrating that ZEB1 activates KIF13A-dependent endocytic recycling to promote front-rear axis polarity and motility. Data are presented as mean values ± SEM, or as mean values ± SD (**j-l**); P values, two-tailed Student's t test, or one-way ANOVA (**c, d, f-l**). R values, Spearman (**a, b**).

**Reporting summary**. Further information on research design is available in the Nature Research Reporting Summary linked to this article.

## Data availability

All data associated with this study are present in the paper and the Supplementary Information. Source data are provided with this paper. RNA-seq data that support the findings of this study have been deposited in Gene Expression Omnibus (accession number GSE102337). Source data are provided with this paper.

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

## Acknowledgements

We thank Dr. Sandra Schmid for valuable scientific insight and Fengju Chen for technical assistance. *Funding*: This work was supported by the National Institutes of Health (NIH) through R01 CA181184 (to J.M.K.), R01 CA2111125 (to J.M.K.), P30 CA125123 (to C.J.C.), K99 CA225633 (to H.F.G.), NIH Lung Cancer SPORE grant P50 CA70907 (to J.M.K.), and Lung Cancer Research Foundation FP#00005299 (to X.T.). J.M.K. holds the Gloria Lupton Tennison Distinguished Endowed Professorship in Lung Cancer. The work was also supported by the generous philanthropic contributions to The University of Texas MD Anderson Lung Cancer Moon Shots Program.

## Author contributions

P.B. conceived, designed, executed, and interpreted all cell culture, labeling, and microscopy experiments. X.T. performed qPCR, Transwell, Boyden chamber, and 3D culture experiments. G.-Y.X. assessed MET trafficking kinetics, quantified endocytosis of biotinylated proteins, and assisted P.B. with image analysis. G.-Y.X., L.S., J.Y. and N.B.-R. assisted X.T. with qPCR experiments. V.Z. assisted P.B. with the analysis of confocal microscopic images. H.-F.G., L.S., X.L. and J.Y. assisted P.B. with cell culture. W.K.R. directed and interpreted mass spectrometry experiments. C.J.C., L.D. and J.R. directed and interpreted bioinformatic analyses. J.M.K. conceived and supervised the project and contributed to the design and interpretation of all experiments.

## Competing interests

J.M.K. has received consulting fees from Halozyme. P.B. has received consulting fees from ExpertConnect. The remaining authors declare no competing interests.
