## [Peer Review File · Nature Communications]

REVIEWER COMMENTS

Reviewer #1 (Remarks to the Author):

In the submitted manuscript the authors have investigated how EMT linked transcription factor ZEB1 influences cellular membrane traffic. They have identified previously that a ZEB1 silences specific microRNAs in lung adenocarcinoma cells and that this influences tumor cell migration and metastasis in EMT-driven lung adenocarcinoma models through a mechanism that involves golgi-mediated vesicular traffic (Tan et al., 2017 J. Clin Invest). In this study they investigate carefully and thoroughly the effect of ZEB1 expression and EMT on endocytosis and recycling of transferrin receptor (TfR) and find that ZEB1 expression shifts the receptor trafficking balance to favor perinuclear vesicle accumulation of TfR. They identify kinesin 13A and clathrin AP1 adaptor subunit, as ZEB1 downregulated mRNAs and characterize that loss of KIF13A functionally contributes to the shifted receptor trafficking balance presumably due to reduced receptor delivery to the plasma membrane. They go on to profile other proteins affected by ZEB1 and identify ITGA5 integrin and MMP14. The focus of the manuscript is fully shifted to ITGA5. The authors suggest that accelerated kinesin mediated recycling of the cellular fibronectin receptor ITGA5 regulates focal adhesions, front-rear polarity and cell migration potentially contributing to EMT phenotype.

There are three major issues with this study.

1) The authors are somewhat playing down the existing knowledge regarding EMT, vesicle traffic and cell front-rear polarity regulation. I understand this to a degree, as unfortunately the strong emphasis on novelty by journal editors, encourages authors to play down earlier work from others as well as their own earlier studies. In my mind the authors should expand their introduction to fix this as I think this makes their data more interesting, not less interesting, and helps the reader in putting these data into context without the need to spend a long time reading the cited references. Please see below for more specific suggestions.

2) This study has clearly started with careful analyses of the link between ZEB1/EMT and vesicular traffic of cargo along the classical well-established clathrin-mediated endocytosis route using TfR as the reporter cargo. This part of the study (Figures 1-4) are carefully conducted and report an interesting and convincing finding that ZEB1 targets KIF13A and AP1S2 regulate the steady-state balance of vesicular traffic in cells favoring prolonged retention of cargo in a perinuclear compartment that is more acidic. This could, in my mind, have several really interesting effects linked to EMT such as prolonged endosomal signaling of EMT-regulation RTKs (MET, AXL, FAK), influence on late endosomal/lysosomal function etc. However, the authors have chosen to downplay the potential generality of vesicle traffic rewiring and rather curiously TfR is not even mentioned in the abstract. Instead they have jumped on one of the hits from the mass-spec screen (which highlighted hundreds of proteins) ITGA5. In the rather poor quality data (compared to the start of the paper which is excellent) they suggest that ITGA5 traffic would be the main contributor to ZEB1 induced cell migration. I don't think this is a valid conclusion. There are certainly effects on ITGA5, but also several other integrins and many other potential adhesion regulators seem to be influenced by ZEB1 according to the author's data and these are not investigated.

3) It is somewhat alarming that the authors are not discussing their ITGA5 linked cell motility data in light of their earlier work on how ZEB1 regulates golgi-mediated vesicle traffic and how this relates

to their earlier interesting finding regarding the role of the ZEB1 regulated Golgi scaffolding protein PAQR11 in cancer cell migration in these same cell lines (Tan et al., 2017). The paper is vaguely cited, but there is no mention about this obvious connection or the findings of that paper which seems acutely relevant for this study.

Specific issues:

1) the authors write in their introduction: “However, the way in which EMT governs endocytosis and intracellular transport of PM-bound proteins remains unclear.” This seems somewhat of an overstatement given that one of the most well established EMT proteins, vimentin, has been implicated in numerous studies to contribute to cell migration, increased front-rear polarity, cell migration and vesicular traffic (including integrin recycling). The authors are encouraged to review the literature in more detail and mention this work in their manuscript. See for examples these papers: (Role of Intermediate Filaments in Vesicular Traffic. Margiotta A, Bucci C. *Cells*. 2016 Apr 25;5(2):20. doi: 10.3390/cells5020020.; Regulation of cell adhesion to collagen via beta1 integrins is dependent on interactions of filamin A with vimentin and protein kinase C epsilon. doi: 10.1016/j.yexcr.2010.02.007.)

2) The authors also write: “Here, we postulated that transcriptional programs governed by EMT activating transcription factors control the endocytosis and intracellular trafficking of PM-bound proteins that regulate cancer cell polarity and motility.” The authors should mention clearly their own previous work where they have already established that ZEB1 mediated regulation of micro-RNAs influence cellular vesicle traffic, cell polarity and migration through influencing Golgi-compactation and vesicle traffic of PM-bound proteins. (Tan et al., *J Clin Invest*. 2017 Jan 3; 127(1): 117–131. ZEB1 regulates anterograde vesicle trafficking to the plasma membrane through silencing miR-200c, miR-148a, and miR-206). They should also cite already published studies where the role of EMT transcriptional programs in regulating cancer cell polarity and migration have been studied, in particular the literature on vimentin and cell migration. For examples references in this review (Cytoskeletal Crosstalk in Cell Migration. doi: 10.1016/j.tcb.2020.06.004.) or primary articles such as (Vimentin regulates EMT induction by Slug and oncogenic H-Ras and migration by governing Axl expression in breast cancer. doi: 10.1038/onc.2010.509; Intermediate filaments control collective migration by restricting traction forces and sustaining cell-cell contacts. doi: 10.1083/jcb.201801162.;

Vimentin Intermediate Filaments Template Microtubule Networks to Enhance Persistence in Cell Polarity and Directed Migration. doi: 10.1016/j.cels.2016.11.011.

3) Figure 3. An alternative explanation to the authors data “In ZEB1 gain- and loss-of-function studies, ZEB1 increased pHrodo Green signal intensity (Fig. 3AC, Movie 2), which suggests that ZEB1 accelerated endocytic vesicle maturation.” would be that ZEB1 influences the dynamics of the late endosomal/lysosomal vesicles.

4) Figure 3. The authors have followed toxin cargo as a function of ZEB1 expression and see increased retrograde traffic to the golgi. They go on to conclude: “ Thus, ZEB1

exerts broad control of vesicular trafficking pathway kinetics.” It would be essential to include also other cargo to warrant this claim. What is the impact on traffic of EMT relevant cargo such as MET. This receptor, upon ligand binding to HGF, is endocytosed and predominantly degraded in lysosomes. Is this traffic affected by ZEB1?

5) The biotinylation based endocytosis assay of plasma membrane derived components followed by mass-spec is a nice set-up and has many exciting protein hits. It is curious, however, that many of the hits are intracellular proteins and the authors are not commenting this. It is also really no obvious why they chose ITGA5 as their focus, given that several integrins, including the EMT relevant TGF-beta modifying integrin beta 5 and beta3 were also identified. Furthermore, I am, rather surprised by the notion that in the absence of ZEB1 ITGA5 (or FN for that matter) would not be endocytosed in 15 minutes at all. This would be very much in contradiction with the existing literature in numerous cancer and normal cell lines. The authors may have also misunderstood the review they cite when claiming that “a fibronectin protease (MMP14) that initiates ITG α 5 ligand-binding and endocytosis”. ITGA5 binds fibronectin with high-affinity independently of MMP14 and as with all b1-integrin heterodimers, ITGA5 is constantly endocytosed and recycled in cells either wo ligand in an inactive conformation or with FN-ligand fragments as an active receptor.

6. Figure 6. What are the cell surface expression levels of ITGA5 in the two cell lines? The endocytosis data should be made more quantitative. The line scans are informative to illustrate the relative distribution of the vesicles but total endocytosed integrin signal from multiple cells from at least 3 independent biological experiments should also be included and the signal normalised to total cell surface signal prior to internalisation. Ideally, the antibody based data should be backed up with biotinylation based endocytosis assays as antibodies may influence the integrin kinetics, in particular the recycling. This is especially important as the authors have used an ITGA5 function blocking antibody (ab25251) in their experiments.

7. supplement 6. It would be essential to show the silencing efficacy using the same ITGA5 antibody than the authors have used in the trafficking assays, not only on the mRNA level. This would also enable the authors to validate their antibody specificity.

8. What the rationale of this experiment? “soluble fibronectin 143 reversed the effect of MMP14 neutralization on ITG α 5 recycling (Supplementary Fig. 6A-C)” Could the authors please clarify and also explain how they consider that FN-treatment rescues traffic of ITGA5 in a set-up where the receptor is labelled with a FN-binding blocking function inhibiting antibody. It may also be difficult to distinguish the effect on FN from the effect of manganese addition as it seems that both are included together and the latter is a superactivator of integrins.

9. Figure 7. What is the evidence that the KIF13A-dependent recycling is specific to ligand-bound ITGA5? What is the effect of blocking receptor recycling with primaquine in this context?

10. Figure 8. What is the effect of KIF13A siRNA on steady-state cell surface ITGA5 levels? Are the other “hit” integrins ITGA2, ITGB3 or ITGB5 affected? Overall, since the data do not indicate that the ZEB1-induced trafficking regulation would be specific to ITGA5, it would be important to investigate other integrins as well and maybe even other cargo.

11. The focal adhesion data in figure 8 would benefit from having images of the FA phenotypes in addition to the quantification. Does KIF13A-silencing influence the subcellular localisation of the FAs?

12. Figure 8H,I. Why is the recycled ITGA5 not re-endocytosed into the cell but accumulates on the PM at 1h?

Reviewer #2 (Remarks to the Author):

The manuscript by Banerjee et al. addresses the mechanisms of protein trafficking involved in the establishment of a front-rear polarity axis during epithelial to mesenchymal transition. The authors demonstrate that the ZEB1 EMT-inducing transcription factor silences microRNAs that target vesicular trafficking regulators and accelerates endocytic recycling, thus facilitating focal adhesion dynamics, front-rear axis polarization and cellular motility.

The manuscript presents an elegant approach aiming to study the coordination of front-rear polarity during EMT. Although limited to analyses in a 2D-setting, overall, the experimental approaches are of high quality and the conclusions are convincing.

Minor points listed below may further increase the impact of the manuscript.

1. The authors used a large panel of human and murine epithelial or mesenchymal cell lines. However, the LUAD cell lines used in this study are positioned at either end of the EMT spectrum. It

would be interesting to comment on the involvement of vesicular trafficking at intermediate stages, or partial EMT.

2. Since ZEB1 and ZEB2 play somehow complementary roles in EMT, is there a role for ZEB2 in vesicular trafficking and front-rear polarity axis establishment?

3. ITG α 5 and the ITG α 5 ligand were found to be endocytosed only in ZEB1 overexpressing cells (at a 15 min time point). The ZEB1 gain and loss of function experiment demonstrates that ZEB1 accelerates the endocytic recycling of a fluorescently tagged anti-ITG α 5. Could the authors show the co-localization of fluorescent anti-ITG α 5 antibodies with the Golgi apparatus (data not shown)?

4. All experiments presented in this study were performed in a 2D-setting. Several recent studies underlined the correlation between mechanical forces from the microenvironment and the compression-induced transcriptomic changes, particularly in EMT-related genes. The authors should discuss their findings in the light of a more physiologically relevant model (in vivo models, 3D, or in vitro compression models, for example).

Reviewer #3 (Remarks to the Author):

The manuscript by Banerjee et al. describes the involvement of the transcription factor ZEB1 in the regulation of front-rear polarity axis in lung adenocarcinoma cells. The authors demonstrated that ZEB1 activities accelerate endocytic recycling by silencing miR-200 and potentially few other microRNAs, which otherwise suppress key vesicular trafficking regulators. The findings are novel and rather interesting, and with a potential clinical relevance as well. All statements and conclusions are based on admirable amount of experimental work, however, in some instances it is not easy to evaluate the quality of the performed experiments.

This reviewer was specifically asked to assess the work involving the surface biotinylation/trafficking experiments and the consequent mass spectrometric analysis. Although not explicitly stated, it appears that this experiment was done only once, without any replicates. And the comparison among the different conditions was entirely based on “identified or not” in this single experiment. With n=1 and considering the semi stochastic nature of the employed shotgun proteomics workflow, it is to be expected for many of the observed/reported differences to be just random, by no means specific to one or other experimental condition. To avoid publishing erroneous information and misleading conclusions, the authors should repeat this experiment and show the correlation between the replicates.

List of protein identifications (e.g. as a Suppl Table) should be provided as well, including all the relevant information – peptide sequences, mass errors etc.

It is stated in the methods that “Proteins were separated by 1D gel electrophoresis, and Coomassie-stained bands were excised and subjected to reduction using...”. It is not clear whether only selected bands were analyzed, or the entire gel-lane was cutout and analyzed, which would be the proper way. If only selected bands were analyzed, what was the criteria for the selection and were the equivalent bands from all lanes been measured as well?

Reviewer #4 (Remarks to the Author):

This is really a solid basic science paper. I am afraid this is not well-suited for Nature Communications as this paper may not attract wider audience. This manuscript may be well received in Cell Biology journals.

Reviewer #1

1) The authors are somewhat playing down the existing knowledge regarding EMT, vesicle traffic and cell front-rear polarity regulation. I understand this to a degree, as unfortunately the strong emphasis on novelty by journal editors, encourages authors to play down earlier work from others as well as their own earlier studies. In my mind the authors should expand their introduction to fix this as I think this makes their data more interesting, not less interesting, and helps the reader in putting these data into context without the need to spend a long time reading the cited references. Please see below for more specific suggestions.

This point is well taken. The Introduction should have stated the current paradigm and the knowledge gap that we addressed more clearly. It has been modified as follows.

“Metastatic disease is a poor prognostic feature and the primary cause of death in patients with epithelial cancers¹. Cancer cells detach from the primary tumor and intravasate into vasculature by undergoing an epithelial-to-mesenchymal transition (EMT), which triggers dissolution of epithelial polarity complexes, assembly of vimentin intermediate filaments, and establishment of a front-rear polarity axis defined by a peri-nuclear, compact Golgi organelle and leading and trailing edges enriched in focal adhesions (FAs) and other actin-based cytoskeletal structures that facilitate attachment to extracellular matrix proteins and promote cell motility^{2, 3, 4}. Transitions between epithelial and mesenchymal states require precise spatial and temporal control of protein transport through endocytic recycling and retrograde vesicular trafficking pathways that facilitate plasma membrane (PM) dynamics, including the recycling of FAs from trailing to leading edges of a motile cell^{5, 6, 7}.

EMT is initiated by transcription factor families (e.g., ZEB, SNAIL, TWIST) that silence the expression of epithelial polarity complexes (e.g., E-cadherin, Crumbs, Claudins) and microRNAs (e.g., miR-200 family, miR-34a, miR-206, miR-148a) that target stemness- and motility-inhibiting genes and EMT-activating transcription factors themselves, creating an adaptive, feed-forward regulatory system that controls reversible switching between epithelial and mesenchymal states^{8, 9, 10, 11, 12, 13, 14}. However, the way in which EMT-activating transcription factors govern protein transport through vesicular trafficking pathways to establish a front-rear polarity axis remains unclear. Here, we addressed this question in murine and human lung adenocarcinoma (LUAD) cell lines at distinct positions on the EMT spectrum. LUAD cells classified as ‘epithelial’ have uniformly epithelial gene expression patterns and exhibit low metastatic propensities, while those classified as ‘mesenchymal’ exhibit partial EMT features characterized by bi-phenotypic gene expression patterns (e.g., high CDH1, CDH2, and VIM), a capacity to undergo EMT or the reverse process in response to extracellular cues, and an aggressive metastatic propensity driven by high levels of the EMT-activating transcription factor ZEB1^{2, 15, 16, 17}.”

2) This study has clearly started with careful analyses of the link between ZEB1/EMT and vesicular traffic of cargo along the classical well-established clathrin-mediated endocytosis route using TfR as the reporter cargo. This part of the study (Figures 1-4) are carefully conducted and report an interesting and convincing finding that ZEB1 targets KIF13A and AP1S2 regulate the steady-state balance of vesicular traffic in cells favoring prolonged retention of cargo in a perinuclear compartment that is more acidic. This could, in my mind, have several really interesting effects linked to EMT such as prolonged endosomal signaling of EMT-regulation RTKs (MET, AXL, FAK), influence on late endosomal/lysosomal function etc. However, the authors have chosen to downplay the potential generality of vesicle traffic rewiring and rather curiously TfR is not even mentioned in the abstract.

We agree that the ZEB1/KIF13A-mediated acceleration of endocytic recycling could impact known EMT-regulated RTKs and that broader implications of accelerated transferrin recycling were not adequately stressed in the manuscript. To address this deficiency, we assessed how ZEB1 influences HGF-induced MET endocytosis and the intracellular trafficking of endocytosed MET. Pasted below is that section of the Results.

“Based on the above evidence that ZEB1 accelerates endocytosis of transmembrane receptors and hastens vesicular trafficking through the endocytic recycling and lysosomal pathways, we speculated that ZEB1 influences the intracellular fate of receptor tyrosine kinases that undergo endocytosis following ligand-binding and are either recycled back to the PM or delivered to lysosomes for degradation⁶. To address this possibility, we assessed the intracellular trafficking of MET, which is routed through endocytic recycling and lysosomal pathways with defined kinetics²² and was identified in the biotinylation screen (Supplementary Data 1). We carried out siRNA-mediated ZEB1 depletion studies on a mesenchymal human LUAD cell line (H1299), treated ZEB1-deficient and -replete cells with hepatocyte growth factor (HGF) to initiate MET endocytosis, and imaged the endocytosed MET at multiple time points to quantify its co-localization with early endosomes (EEA1⁺), recycling endosomes (Rab11⁺), and late endosomes/lysosomes (LAMP1⁺). We found that ZEB1 deficiency delayed MET trafficking through the endocytic recycling compartment and prevented MET accumulation in lysosomes (Fig. 5a, b and Supplementary Fig. 3). In line with these findings, HGF-induced MET protein degradation was delayed in ZEB1-deficient cells (Fig. 5c). Thus, ZEB1 drives MET turnover by enhancing HGF-induced MET endocytosis and delivery of endocytosed MET to the lysosomal compartment.”

Additionally, the Abstract and Results sections have been completely reorganized to address these concerns.

3) Instead they have jumped on one of the hits from the mass-spec screen (which highlighted hundreds of proteins) ITGA5. In the rather poor quality data (compared to the start of the paper which is excellent) they suggest that ITGA5 traffic would be the main contributor to ZEB1 induced cell migration. I don't think this is a valid conclusion. There are certainly effects on ITGA5, but also several other integrins and many other potential adhesion regulators seem to be influenced by ZEB1 according to the author's data and these are not investigated.

We agree with the reviewer that ZEB1 is likely to promote motility through multiple adhesion regulators, and we did not mean to imply that ITGA5 was the main contributor. Our intention with the ITGA5 studies was to provide proof-of-principle that ZEB1 accelerates the endocytic recycling of a key focal adhesion component. We chose ITGA5 simply based on its known role as a focal adhesion component and driver of cell motility, but the same argument could have been made for other candidates identified. Given this and other concerns expressed about the ITGA5 data, we have removed the majority of the ITGA5 data from the manuscript.

4) It is somewhat alarming that the authors are not discussing their ITGA5 linked cell motility data in light of their earlier work on how ZEB1 regulates golgi-mediated vesicle traffic and how this relates to their earlier interesting finding regarding the role of the ZEB1 regulated Golgi scaffolding protein PAQR11 in cancer cell migration in these same cell lines (Tan et al., 2017). The paper is vaguely cited, but there is no mention about this obvious connection or the findings of that paper which seems acutely relevant for this study.

We agree that this omission was myopic on our part and have modified the Discussion section to address this deficiency. The first paragraph now reads as follows.

“EMT is a transcriptionally governed process that initiates cell motility by generating a leading edge with actin-based cytoskeletal projections that cycle from back to front in motile cells^{8, 45}. Such dynamics require precise spatial and temporal control of vesicular trafficking. Yet, the way in which EMT-activating transcription factors govern vesicular trafficking networks that effect cell polarity change remains unclear. Here, we show that the EMT activator ZEB1 is a key regulator of endocytic vesicular transport dynamics that establish a front-rear polarity axis and drive cell motility (Fig. 8m). In the context of our prior report that a Golgi scaffolding protein upregulated by ZEB1 activates a Golgi compaction process that drives FA maturation and cell motility², we conclude that a ZEB1 activates a transcriptional program that coordinates Golgi dynamics with accelerated cargo transport to effect cell polarity change”.

5) The authors write in their introduction: “However, the way in which EMT governs endocytosis and intracellular transport of PM-bound proteins remains unclear.” This seems somewhat of an overstatement given that one of the most well established EMT proteins, vimentin, has been implicated in numerous studies to contribute to cell migration, increased front-rear polarity, cell migration and vesicular traffic (including integrin recycling). The authors are encouraged to review the literature in more detail and mention this work in their manuscript. See for examples these papers: (Role of Intermediate Filaments in Vesicular Traffic. Margiotta A, Bucci C. Cells. 2016 Apr 25;5(2):20. doi: 10.3390/cells5020020.; Regulation of cell adhesion to collagen via beta1 integrins is dependent on interactions of filamin A with vimentin and protein kinase C epsilon. doi: 10.1016/j.yexcr.2010.02.007.)

We appreciate the reviewer’s point and have modified the Introduction to incorporate these references and the concepts they convey.

6) The authors also write: “Here, we postulated that transcriptional programs governed by EMT activating transcription factors control the endocytosis and intracellular trafficking of PM-bound proteins that regulate cancer cell polarity and motility.” The authors should mention clearly their own previous work where they have already established that ZEB1 mediated regulation of micro-RNAs influence cellular vesicle traffic, cell polarity and migration through influencing Golgi-compaction and vesicle traffic of PM-bound proteins. (Tan et al., J Clin Invest. 2017 Jan 3; 127(1): 117–131. ZEB1 regulates anterograde vesicle trafficking to the plasma membrane through silencing miR-200c, miR-148a, and miR-206). They should also cite already published studies where the role of EMT transcriptional programs in regulating cancer cell polarity and migration have been studied, in particular the literature on vimentin and cell migration. For examples references in this review (Cytoskeletal Crosstalk in Cell Migration. doi: 10.1016/j.tcb.2020.06.004.) or primary articles such as (Vimentin regulates EMT induction by Slug and oncogenic H-Ras and migration by governing Axl expression in breast cancer. doi: 10.1038/onc.2010.509; Intermediate filaments control collective migration by restricting traction forces and sustaining cell-cell contacts. doi: 10.1083/jcb.201801162.; Vimentin Intermediate Filaments Template Microtubule Networks to Enhance Persistence in Cell Polarity and Directed Migration. doi: 10.1016/j.cels.2016.11.011.

The reviewer has rightly pointed out that ZEB1-silenced microRNAs are a critical link to the transcriptional targets identified here and elsewhere. We have modified the Introduction to make this point and have incorporated literature recommended by the reviewer.

7) Figure 3. An alternative explanation to the authors data “In ZEB1 gain- and loss-of-function studies, ZEB1 increased pHrodo Green signal intensity (Fig. 3AC, Movie 2), which suggests that ZEB1 accelerated endocytic vesicle maturation.” would be that ZEB1 influences the dynamics of the late endosomal/lysosomal vesicles.

This point is on target and quite relevant to our finding that ZEB1 accelerated MET trafficking to lysosomes and HGF-dependent MET degradation. That section of the Results section has been modified as follows.

“Endocytosed cargos can be either recycled to the PM or routed to lysosomes for degradation²⁰. Lysosomal maturation is associated with a luminal acidification process that is critical for lysosomal functions²¹. To determine whether ZEB1 hastens the maturation of early endosomes into lysosomes, we utilized Tfn tagged with pHrodo Green, a fluorescent pH indicator that increases its intensity in vesicles with lower pH. We found that, in ZEB1 gain- and loss-of-function studies, ZEB1 increased pHrodo Green signal intensity (Fig. 4a-c and Supplementary Movie 2), which suggests that ZEB1 promotes endocytic vesicle maturation into lysosomes.

Based on the above evidence that ZEB1 accelerates endocytosis of transmembrane receptors and hastens vesicular trafficking through the endocytic recycling and lysosomal pathways, we speculated that ZEB1 influences the intracellular fate of receptor tyrosine kinases that undergo endocytosis following ligand-binding and are either recycled back to the PM or delivered to lysosomes for degradation⁶. To address this possibility, we assessed the intracellular trafficking of MET, which is routed through endocytic recycling and lysosomal pathways with defined kinetics²² and was identified in the biotinylation screen (Supplementary Data 1). We carried out siRNA-mediated ZEB1 depletion studies on a mesenchymal human LUAD cell line (H1299), treated ZEB1-deficient and -replete cells with hepatocyte growth factor (HGF) to initiate MET endocytosis, and imaged the endocytosed MET at multiple time points to quantify its co-localization with early endosomes (EEA1⁺), recycling endosomes (Rab11⁺), and late endosomes/lysosomes (LAMP1⁺). We found that ZEB1 deficiency delayed MET trafficking through the endocytic recycling compartment and prevented MET accumulation in lysosomes (Fig. 5a, b and Supplementary Fig. 3). In line with these findings, HGF-induced MET protein degradation was delayed in ZEB1-deficient cells (Fig. 5c). Thus, ZEB1 drives MET turnover by enhancing HGF-induced MET endocytosis and delivery of endocytosed MET to the lysosomal compartment.”

8) Figure 3. The authors have followed toxin cargo as a function of ZEB1 expression and see increased retrograde traffic to the golgi. They go on to conclude: “ Thus, ZEB1 exerts broad control of vesicular trafficking pathway kinetics.” It would be essential to include also other cargo to warrant this claim. What is the impact on traffic of EMT relevant cargo such as MET. This receptor, upon ligand binding to HGF, is endocytosed and predominantly degraded in lysosomes. Is this traffic affected by ZEB1?

As stated above, we found that ZEB1 accelerated HGF-induced MET endocytosis and trafficking of endocytosed MET to lysosomes.

9) The biotinylation based endocytosis assay of plasma membrane derived components

followed by mass-spec is a nice set-up and has many exciting protein hits. It is curious, however, that many of the hits are intracellular proteins and the authors are not commenting this.

We speculate that intracellular proteins formed complexes with endocytosed biotinylated proteins before cell lysis and were immunoprecipitated with the biotinylated proteins. We have added this point to the Results section.

10) It is also really no obvious why they chose ITGA5 as their focus, given that several integrins, including the EMT relevant TGF-beta modifying integrin beta 5 and beta3 were also identified.

As stated above, our intention with the ITGA5 studies was to provide proof-of-principle that ZEB1 accelerates the endocytic recycling of a key focal adhesion component. We chose ITGA5 based on its known role as a focal adhesion component and driver of cell motility, but the same argument could have been made for other candidates identified. Given that the reviewer expressed this and other concerns about the ITGA5 data, we have removed the bulk of the ITGA5 data from the manuscript.

11) Furthermore, I am, rather surprised by the notion that in the absence of ZEB1 ITGA5 (or FN for that matter) would not be endocytosed in 15 minutes at all. This would be very much in contradiction with the existing literature in numerous cancer and normal cell lines.

As discussed below in response to point #13, Western blot analysis did detect ITGA5 and FN endocytosis in control cells at 15 min, suggesting that the level of ITGA5 endocytosis in control cells at 15 min was beneath the detection limits of our microscopy conditions.

12) The authors may have also misunderstood the review they cite when claiming that “a fibronectin protease (MMP14) that initiates ITG α 5 ligand-binding and endocytosis”. ITGA5 binds fibronectin with high-affinity independently of MMP14 and as with all b1-integrin heterodimers, ITGA5 is constantly endocytosed and recycled in cells either without ligand in an inactive conformation or with FN-ligand fragments as an active receptor.

This sentence was deleted when the ITGA5 data were removed from the manuscript.

13) Figure 6. What are the cell surface expression levels of ITGA5 in the two cell lines? The endocytosis data should be made more quantitative. The line scans are informative to illustrate the relative distribution of the vesicles but total endocytosed integrin signal from multiple cells from at least 3 independent biological experiments should also be included and the signal normalised to total cell surface signal prior to internalisation.

To more quantitatively assess endocytosis of ITGA5 and other proteins identified by LC-MS analysis, we repeated the biotinylation experiment and quantified each endocytosed protein as a percentage of initial surface-bound protein. Biotin-labelled 393P_ZEB1 cells and 393P_vector cells were placed on ice (T=0), transferred to 37°C to initiate endocytosis, and lysed after 15 min. Lysates were subjected to pull-down with streptavidin beads, and western blot analysis of ITGA5 and other proteins identified by LC-MS was performed on the streptavidin-bound proteins. The experiment was performed 3 times, and densitometric values were averaged from triplicate samples, which showed that a higher percentage of surface-bound ITGA5 was endocytosed in 393P_ZEB1 cells than 393P_vector cells (Fig. 1f).

14) Ideally, the antibody based data should be backed up with biotinylation based endocytosis assays as antibodies may influence the integrin kinetics, in particular the recycling. This is especially important as the authors have used an ITGA5 function blocking antibody (ab25251) in their experiments.

The reviewer has raised an important point. We cannot exclude the possibility that the ITGA5 antibody influenced ITGA5 recycling kinetics. Although the biotinylation-based endocytosis assay described above (#13) showed that a higher percentage of surface-bound ITGA5 was endocytosed in 393P_ZEB1 cells than 393P_vector cells, we decided to remove all of the antibody-based ITGA5 labeling studies from the paper to avoid any incorrect conclusions due to confounding effects of the antibody on ITGA5 trafficking kinetics.

15) supplement 6. It would be essential to show the silencing efficacy using the same ITGA5 antibody than the authors have used in the trafficking assays, not only on the mRNA level. This would also enable the authors to validate their antibody specificity.

As suggested, we performed western blot analysis and showed that ITGA5 siRNA depletes ITGA5 protein levels.

16) What the rationale of this experiment? “soluble fibronectin 143 reversed the effect of MMP14 neutralization on ITGA5 recycling (Supplementary Fig. 6A-C)” Could the authors please clarify and also explain how they consider that FN-treatment rescues traffic of ITGA5 in a set-up where the receptor is labelled with a FN-binding blocking function inhibiting antibody. It may also be difficult to distinguish the effect on FN from the effect of manganese addition as it seems that both are included together and the latter is a superactivator of integrins. Figure 7. What is the evidence that the KIF13A-dependent recycling is specific to ligand-bound ITGA5?

Given concerns about potentially confounding effects of the anti-ITGA5 antibody and manganese treatment on ITGA5 recycling kinetics, we have removed Figure 7 and Supplementary Figure 6 from the paper.

18) What is the effect of blocking receptor recycling with primaquine in this context?

Removal of the ITGA5 antibody labeling studies obviates the need for confirmatory studies with primaquine.

19) Figure 8. What is the effect of KIF13A siRNA on steady-state cell surface ITGA5 levels? Are the other “hit” integrins ITGA2, ITGB3 or ITGB5 affected? Overall, since the data do not indicate that the ZEB1-induced trafficking regulation would be specific to ITGA5, it would be important to investigate other integrins as well and maybe even other cargo.

We removed ITGA5 trafficking studies from the paper. In the revised manuscript, KIF13A depletion studies are restricted to the transferrin and cholera toxin B trafficking experiments.

20) The focal adhesion data in figure 8 would benefit from having images of the FA phenotypes in addition to the quantification. Does KIF13A-silencing influence the subcellular localisation of the FAs?

We have added representative images of focal adhesions. Subcellular localization of focal adhesions was not formally assessed.

21) Figure 8H,I. Why is the recycled ITGA5 not re-endocytosed into the cell but accumulates on the PM at 1h?

ITGA5 tracking studies were removed from the paper.

Reviewer #2

1) The authors used a large panel of human and murine epithelial or mesenchymal cell lines. However, the LUAD cell lines used in this study are positioned at either end of the EMT spectrum. It would be interesting to comment on the involvement of vesicular trafficking at intermediate stages, or partial EMT.

The language we used to describe the LUAD cell lines was unintentionally misleading. In fact, the cells classified as mesenchymal exhibit a partial EMT. We should have been more precise, particularly given that a growing body of literature shows that partial EMT is uniquely associated with enhanced metastatic activity. We have corrected that language. The portion of the Introduction that addresses the LUAD cell lines now reads as follows.

“EMT is initiated by transcription factor families (e.g., ZEB, SNAIL, TWIST) that silence the expression of epithelial polarity complexes (e.g., E-cadherin, Crumbs, Claudins) and microRNAs (e.g., miR-200 family, miR-34a, miR-206, miR-148a) that target stemness- and motility-inhibiting genes and EMT-activating transcription factors themselves, creating an adaptive, feed-forward regulatory system that controls reversible switching between epithelial and mesenchymal states^{8, 9, 10, 11, 12, 13, 14}. However, the way in which EMT-activating transcription factors govern protein transport through vesicular trafficking pathways to establish a front-rear polarity axis remains unclear. Here, we addressed this question in murine and human lung adenocarcinoma (LUAD) cell lines at distinct positions on the EMT spectrum. LUAD cells classified as ‘epithelial’ have uniformly epithelial gene expression patterns and exhibit low metastatic propensities, while those classified as ‘mesenchymal’ exhibit partial EMT features characterized by bi-phenotypic gene expression patterns (e.g., high CDH1, CDH2, and VIM), a capacity to undergo EMT or the reverse process in response to extracellular cues, and an aggressive metastatic propensity driven by high levels of the EMT-activating transcription factor ZEB1^{2, 15, 16, 17}.”

2) Since ZEB1 and ZEB2 play somehow complementary roles in EMT, is there a role for ZEB2 in vesicular trafficking and front-rear polarity axis establishment?

Although we have not addressed the role of ZEB2 in our model, ZEB1 and ZEB2 do have partially overlapping transcriptional targets, so it is entirely possible that ZEB2 influences vesicular trafficking and front-rear cell polarity establishment. Our findings are meant to provide proof-of-principle that an EMT-activating transcription factor has the capacity to govern vesicular trafficking and to set the stage for future studies on model systems driven by ZEB2 and other EMT-activating transcription factors.

3) ITGα5 and the ITGα5 ligand were found to be endocytosed only in ZEB1 overexpressing cells (at a 15 min time point). The ZEB1 gain and loss of function experiment demonstrates that ZEB1 accelerates the endocytic recycling of a fluorescently tagged anti-ITGα5. Could the authors show the co-localization of fluorescent anti-ITGα5 antibodies with the Golgi apparatus (data not shown)?

Given concerns expressed by reviewer 1 about potentially confounding effects of the fluorescently-tagged ITGA5 antibody on ITGA5 trafficking, we have removed these findings from the paper.

4) All experiments presented in this study were performed in a 2D-setting. Several recent studies underlined the correlation between mechanical forces from the microenvironment and the compression-induced transcriptomic changes, particularly in EMT-related genes. The authors should discuss their findings in the light of a more physiologically relevant model (in vivo models, 3D, or in vitro compression models, for example).

To address this question, we assessed how KIF13A depletion influenced the capacity of 393P_ZEB1 cells to generate invasive multicellular aggregates in 3-dimensional collagen and found that KIF13A depletion reduced the formation of collectively invasive protrusions and single invasive cells (Fig. 8I).

Reviewer #3

1) This reviewer was specifically asked to assess the work involving the surface biotinylation/trafficking experiments and the consequent mass spectrometric analysis. Although not explicitly stated, it appears that this experiment was done only once, without any replicates. And the comparison among the different conditions was entirely based on “identified or not” in this single experiment. With n=1 and considering the semi stochastic nature of the employed shotgun proteomics workflow, it is to be expected for many of the observed/reported differences to be just random, by no means specific to one or other experimental condition. To avoid publishing erroneous information and misleading conclusions, the authors should repeat this experiment and show the correlation between the replicates.

To address this concern, we quantified the fractions of surface-bound proteins that were endocytosed by carrying out Western blot analysis on biotinylated proteins recovered from 393P_vector cells and 393P_ZEB1 cells lysed prior to or 15 min after initiating endocytosis. The experiment was repeated twice to obtain triplicate biological samples at each timepoint. Densitometric values from the triplicate samples were averaged, which showed that proteins identified by LC-MS were endocytosed to a greater extent in 393P_ZEB1 cells than 393P_vector cells (Fig. 1f). That section of the Results now reads as follows.

“ZEB1 enhances endocytosis of a broad range of transmembrane receptors. We initially carried out a surface biotinylation assay on an epithelial LUAD cell line (393P) that undergoes EMT in response to ectopic ZEB1 expression¹⁸. Cells (393P_ZEB1 or 393P_vector) were treated on ice with a cleavable and membrane-impermeable biotin moiety. After transferring to 37⁰ C to initiate endocytosis, cells were washed to remove residual PM-bound biotinylated proteins, fixed, and stained with fluorescent streptavidin, which showed that ectopic ZEB1 expression accelerated the endocytosis and intracellular transport of biotinylated proteins to a peri-nuclear region that co-localized with the Golgi (Fig. 1a-c). To identify the PM-bound proteins that were endocytosed, the cells were treated with biotin and lysed prior to or 15 min after initiating endocytosis. Endocytosed biotinylated proteins were recovered from the lysates, resolved using SDS-PAGE, and subjected to liquid chromatography-mass spectrometry (LC-MS). Sequencing identified 966 proteins (≥ 2 peptides per protein, $<1\%$ false discovery rate using Percolator and a decoy database), 132 of which were present at baseline on the PM of

both cells but were endocytosed only in 393P_ZEB1 cells (Fig. 1d). By Ingenuity pathway analysis, these proteins are involved in diverse cellular functions (Supplementary Data 1), and they are components of an EMT-associated gene expression signature in a human LUAD cohort from The Cancer Genome Atlas (TCGA) (Fig. 1e). The proteins identified include, among others, a diverse array of transmembrane receptors (e.g., MET, PLXNA1, PLXNB2, ITGA5, ITGB3, EPHA4, EGFR, ERBB2, IGFIR, BMPR2), their cognate ligands (e.g., SEMA3C, LAMC1, FN1), and cytoplasmic proteins (e.g., MYH10, HSP90, FLNB) that presumably co-immunoprecipitated with biotinylated PM-bound proteins (Supplementary Data 1). To quantify the fractions of surface-bound proteins that were endocytosed, Western blot analysis was carried out on biotinylated proteins recovered from 393P_vector cells and 393P_ZEB1 cells lysed prior to or 15 min after initiating endocytosis, which showed that proteins identified by LC-MS were endocytosed to a greater extent in 393P_ZEB1 cells than 393P_vector cells (Fig. 1f).”

2) List of protein identifications (e.g. as a Suppl Table) should be provided as well, including all the relevant information – peptide sequences, mass errors etc.

Protein analytics (peptide sequences, mass errors) from the surface biotinylation experiment are in Supplementary Data 2, which is called out in the Methods section.

3) It is stated in the methods that “Proteins were separated by 1D gel electrophoresis, and Coomassie-stained bands were excised and subjected to reduction using....”. It is not clear whether only selected bands were analyzed, or the entire gel-lane was cutout and analyzed, which would be the proper way. If only selected bands were analyzed, what was the criteria for the selection and were the equivalent bands from all lanes been measured as well?

The gel was run until the dye front was about 1cm into the separating gel. We then cut out that 1 cm lane and sliced it into ~2mm slices, digested each separately and ran each by LC-MS separately. Then we combined the raw data into one file for purposes of database searching. We did NOT only select certain bands; the entire 1cm was analyzed. This is a typical workflow for MUDPIT (multidimensional protein identification technology). It is also known as GeLC-MS.

REVIEWER COMMENTS

Reviewer #1 (Remarks to the Author):

The authors have addressed all my remarks and re-focused the study as suggested

Reviewer #2 (Remarks to the Author):

The authors have satisfactorily responded to all my questions and made the necessary changes to the manuscript.

Reviewer #3 (Remarks to the Author):

The authors did very little to address the major concern related to the validity of the 132 proteins reported here as being endocytosed by ZEB1.

Authors were explicitly asked to repeat the biotinylation/mass spec experiment and show the correlation between the replicates. Yet, the experiment is still n=1 in the revised manuscript. It is still to be expected for many of the reported 132 candidates to be just randomly identified in one sample and not the other, thereby wrongly reported as regulated by ZEB1.

Reviewer #3:

The authors did very little to address the major concern related to the validity of the 132 proteins reported here as being endocytosed by ZEB1. Authors were explicitly asked to repeat the biotinylation/mass spec experiment and show the correlation between the replicates. Yet, the experiment is still n=1 in the revised manuscript. It is still to be expected for many of the reported 132 candidates to be just randomly identified in one sample and not the other, thereby wrongly reported as regulated by ZEB1.

Our western blot analysis of proteins identified in the biotinylation screen validated that ZEB1 accelerates endocytosis of receptor tyrosine kinases. Therefore, we feel that repeating the biotinylation and LC-MS analysis is unnecessary. However, to address this concern, we have removed the proteomic analysis of endocytosed biotinylated proteins from the revised manuscript. These changes do not diminish the strength of our argument that endocytic vesicular trafficking is transcriptionally governed to execute cell polarity change.